# Generalization Bounds via Conditional $f$-Information

**Ziqiao Wang**
School of Computer Science and Technology
Tongji University
Shanghai, China
`ziqiaowang@tongji.edu.cn`

**Yongyi Mao**
School of Electrical Engineering and Computer Science
University of Ottawa
Ottawa, Canada
`ymao@uottawa.ca`

## Abstract

In this work, we introduce novel information-theoretic generalization bounds using the conditional $f$-information framework, an extension of the traditional conditional mutual information (MI) framework. We provide a generic approach to derive generalization bounds via $f$-information in the supersample setting, applicable to both bounded and unbounded loss functions. Unlike previous MI-based bounds, our proof strategy does not rely on upper bounding the cumulant-generating function (CGF) in the variational formula of MI. Instead, we set the CGF or its upper bound to zero by carefully selecting the measurable function invoked in the variational formula. Although some of our techniques are partially inspired by recent advances in the coin-betting framework (e.g., Jang et al. (2023)), our results are independent of any previous findings from regret guarantees of online gambling algorithms. Additionally, our newly derived MI-based bound recovers many previous results and improves our understanding of their potential limitations. Finally, we empirically compare various $f$-information measures for generalization, demonstrating the improvement of our new bounds over the previous bounds.

## 1 Introduction

Understanding generalization is a longstanding topic of machine learning research. Recently, information-theoretic generalization measures, e.g., mutual information (MI) between the input (i.e. data) and output (i.e. hypothesis) of a learning algorithm, have been proposed by [2–4]. While original MI-based information-theoretic generalization bounds have made significant progress in analyzing the generalization of stochastic gradient-based algorithms in nonconvex learning problems (e.g., deep learning) [5–13], they are found to have several limitations. One of the most severe limitations is their unboundedness, where the MI measure can be infinite while the true generalization error is small [14]. To mitigate this issue, various bound-tightening techniques have been proposed [15, 6, 7]. Among them, [16] introduces a conditional mutual information (CMI) framework, which allows for the derivation of a CMI-based bound that is strictly upper-bounded by a constant. In the CMI framework, an additional "ghost sample" is introduced alongside the original sample drawn from the unknown data distribution. A Bernoulli random variable sequence is then used to determine the membership of training data from these two samples. Ultimately, the CMI between these Bernoulli random variables and the output hypothesis of the learning algorithm, conditioned on these two samples, characterizes generalization. This setup, known as the "supersample" setting, ensures the

38th Conference on Neural Information Processing Systems (NeurIPS 2024).

boundedness of the CMI bound due to the constant entropy of a Bernoulli random variable. Intuitively, this CMI quantity leaks less information between data and the hypothesis, resulting in tighter bounds compared to the original MI quantity, as theoretically justified in [17]. Furthermore, several studies have aimed to further tighten CMI-based bounds. For example, previous tightening techniques in [15, 6, 7] are also applicable to CMI bounds [17–21]. Notably, another way to tighten the CMI bound is by utilizing variants of the original CMI quantity, such as functional CMI ($f$-CMI) [22], evaluated CMI (e-CMI) [16, 23], and loss-difference CMI (ld-CMI) [24]. As the output hypothesis is replaced by the predictions, loss pairs, and loss differences of the two samples, these CMI quantities are tighter than the original hypothesis-based CMI due to the data-processing inequality. While most information-theoretic bounds in the supersample setting focus on (conditional) mutual information measures, it is natural to ask whether there is a general way to extend this CMI framework to other statistical distance/divergence measures, such as other $f$-divergences.

Using alternative dependence measures to replace MI in information-theoretic generalization bounds has been studied in several existing works [25–30], with some of them considering the supersample setting. In particular, [26] provides some Wasserstein distance-based and total variation-based bounds in the supersample setting. Furthermore, [29] proposes a general convex analysis-based framework that replaces the input-output MI quantity in [2–4] with any strongly convex function. While [29] itself does not discuss the supersample setting, the further extension in [30] indeed considers invoking the supersample setting to strengthen the bounds.

In this work, we present a generic approach to derive generalization bounds based on conditional $f$-information, a natural extension from mutual information to other $f$-dvergence-based dependence measures. Our proof strategy is significantly different from the original CMI framework [16] and the convex analysis (or regret analysis) based framework [29, 30]. Specifically, our development starts from the variational representation of $f$-divergence, which involves the convex conjugate of a convex function used to define the $f$-divergence. While the previous CMI framework focuses on upper-bounding the cumulant generating function (CGF) of a selected measurable function (which relates to the generalization error), we make a particular choice of such a measurable function, namely the inverse of that convex conjugate function. This choice makes CGF equal to (or upper bounded by) zero, hence eliminated from the variational representation. The remaining task is to lower bound this inverse of the convex conjugate function under a joint distribution. In the case of MI (or KL), the expectation of this inverse function is close to the *log wealth* quantity used in the coin-betting framework for obtaining concentration inequalities [31, 1, 32]. Indeed, the lower bounding techniques here are inspired by some inequalities used in the coin-betting framework [31, 1, 32], and we extend them to the cases of other $f$-divergence in the supersample setting. Notably, unlike [30, 1], our conditional $f$-information generalization bounds do not rely on any existing regret guarantees of online learning algorithms. We discuss the connection with the coin-betting framework in Appendix E.

Given that $f$-divergences all obey the data-processing inequality [33], we focus on using loss difference (ld)-based $f$-information, an extension of ld-CMI in [24], in this work. Specifically, our main contributions are summarized as follows: 1) In the case where the loss difference between a data pair is bounded, we first provide a novel variational formula (cf. Lemma 3.1) for $f$-information, enabling us to derive conditional $f$-information-based generalization bounds; 2) For the KL case, we first provide an "oracle" CMI bound (cf. Theorem 3.1). This bound recovers many previous CMI bounds, including the square-root bound and fast-rate bounds in low empirical risk settings. It also helps us understand the potential looseness of previous square-root CMI bounds, where some quantities that can vanish as the sample size $n$ increases are simply ignored. Additionally, some novel fast-rate bounds are implied (cf. Corollary 3.1-3.2). Particularly, Corollary 3.2 contains a term in the form of the product of total variation and KL divergence, a similar quantity also appearing in a recent PAC-Bayesian bound [34]; 3) We present several other $f$-information-based bounds, including the looser measure, $\chi^2$-information (cf. Theorem B.2) and tighter measures, squared Hellinger (SH)-information (cf. Theorem 3.2) and Jensen-Shannon (JS)-information (cf. Theorem 3.3). Due to the similarity in expressions, most arguments for our CMI bounds are also applicable to these bounds based on $f$-information measures; 4) We extend our framework to the unbounded loss difference case, where we provide a refined variational formula for $f$-information (cf. Lemma 4.1), enabling us to provide a novel $f$-information-based bound (cf. Theorem 4.1 for the case of KL). The obtained bound is no worse than a previous information-theoretic generalization bound in [25] (adapted to the same supersample setting); 5) Empirical results show that the new bounds in our framework, particularly the squared Hellinger-information bound, outperform previous results.

## 2 Preliminaries

**Notation**  Throughout this paper, we adopt a convention where capitalized letters denote random variables, and their corresponding lowercase letters represent specific realizations. Let $P_X$ denote the distribution of a random variable $X$, and $P_{X|Y}$ represent the conditional distribution of $X$ given $Y$. When conditioning on a specific realization, we use $P_{X|Y=y}$ or simply $P_{X|y}$. Additionally, we use $\mathbb{E}_X$ and $\mathbb{E}_P$ interchangeably to denote the expectation over $X \sim P$, whenever it is clear from context. Moreover, $\mathbb{E}_{X|Y=y}$ (or $\mathbb{E}_{X|y}$) denotes the expectation over $X \sim P_{X|Y=y}$.

**Generalization Error**  Let $\mathcal{Z}$ be the domain of the instances, and let $\mathcal{W}$ be the domain of the hypotheses. We denote the unknown distribution of the instances by $\mu$, and let $S = \{Z_i\}_{i=1}^n \overset{i.i.d}{\sim} \mu$ be the training sample. A learning algorithm $\mathcal{A}$, characterized by a conditional distribution $P_{W|S}$, takes the training sample $S$ as the input and outputs a hypothesis $W \in \mathcal{W}$, i.e. $\mathcal{A} : \mathcal{Z}^n \to \mathcal{W}$. To evaluate the quality of the output hypothesis $W$, we use a loss function $\ell : \mathcal{W} \times \mathcal{Z} \to \mathbb{R}_0^+$. For any hypothesis $w$, the population risk is defined as $L_\mu(w) \triangleq \mathbb{E}_{Z'}[\ell(w, Z')]$, where $Z' \sim \mu$ is an independent testing instance. In practice, since $\mu$ is unknown, we use the empirical risk, which is defined as $L_S(w) \triangleq \frac{1}{n} \sum_{i=1}^n \ell(w, Z_i)$, as a proxy for the population risk of $w$. The expected generalization error is then defined as $\mathcal{E}_\mu(\mathcal{A}) \triangleq \mathbb{E}_W[L_\mu(W)] - \mathbb{E}_{W,S}[L_S(W)]$.

**Supersample Setting**  We follow the traditional supersample setting introduced by [16]. Let an $n \times 2$ matrix $\widetilde{Z} \in \mathcal{Z}^{n \times 2}$ be the supersample, where each entry is drawn i.i.d. from the data distribution $\mu$. We index the columns of $\widetilde{Z}$ by $\{0, 1\}$, and denote the $i$th row of $\widetilde{Z}$ as $\widetilde{Z}_i$ with entries $(\widetilde{Z}_{i,0}, \widetilde{Z}_{i,1})$. We use $U = \{U_i\}_{i=1}^n \sim \text{Unif}(\{0,1\}^n)$, independent of $\widetilde{Z}$, to determine the training sample membership from $\widetilde{Z}$. Specifically, when $U_i = 0$, $\widetilde{Z}_{i,0}$ in $\widetilde{Z}$ is included in the training set $S$, and $\widetilde{Z}_{i,1}$ is used for testing; $U_i = 1$ dictates the opposite case. Let $\overline{U}_i = 1 - U_i$, the training sample $S$ is then equivalent to $\widetilde{Z}_U = \{\widetilde{Z}_{i,U_i}\}_{i=1}^n$, and the testing sample is $\widetilde{Z}_{\overline{U}} = \{\widetilde{Z}_{i,\overline{U}_i}\}_{i=1}^n$. Additionally, we let $L_{i,0} \triangleq \ell(W, \widetilde{Z}_{i,0})$ and define $L_{i,1}$ similarly. Let $\Delta L_i = L_{i,1} - L_{i,0}$ be the loss difference in the $i$th row of $\widetilde{Z}$. To avoid complicated subscripts, we also use superscripts $+$ and $-$ to replace the subscripts $0$ and $1$, respectively, namely, $\widetilde{Z}_i^+ = \widetilde{Z}_{i,0}$, $\widetilde{Z}_i^- = \widetilde{Z}_{i,1}$, $L_i^+ = L_{i,0}$ and $L_i^- = L_{i,1}$. Furthermore, we define $G_i \triangleq (-1)^{U_i} \Delta L_i$, i.e. the testing loss minus the training loss at position $i$. Clearly, $\frac{1}{n} \sum_{i=1}^n G_i$ is an unbiased estimator of $\mathcal{E}_\mu(\mathcal{A})$. In other words, $\mathcal{E}_\mu(\mathcal{A}) = \frac{1}{n} \sum_{i=1}^n \mathbb{E}_{\Delta L_i, U_i}[G_i]$.

**$f$-Divergence and $f$-Information**  The family of $f$-divergence is defined as follows.

**Definition 2.1** ($f$-divergence [33])**.** *Let $P$ and $Q$ be two distributions on $\Theta$, and let $\phi : \mathbb{R}_+ \to \mathbb{R}$ be a convex function with $\phi(1) = 0$. If $P \ll Q$, then the $f$-divergence between $P$ and $Q$ is defined as*

$$\mathrm{D}_\phi(P||Q) \triangleq \mathbb{E}_Q\left[\phi\left(\frac{dP}{dQ}\right)\right],$$

*where $\frac{dP}{dQ}$ is a Radon-Nikodym derivative.*

A variational representation for $f$-divergence, as provided below, has been independently investigated in previous works [35, 25, 36, 37, 33]. Recently, this variational representation has also been applied to PAC-Bayesian generalization theory [38] and domain adaptation theory [39, 40].

**Lemma 2.1** ([37, Corollary 3.5])**.** *Let $\phi^*$ be the convex conjugate[1] of $\phi$, and $\mathcal{G} = \{g : \Theta \to \text{dom}(\phi^*)\}$. The following, known as variational formula of $f$-divergence, holds:*

$$\mathrm{D}_\phi(P||Q) = \sup_{g \in \mathcal{G}} \mathbb{E}_{\theta \sim P}[g(\theta)] - \inf_{\lambda \in \mathbb{R}}\{\mathbb{E}_{\theta \sim Q}[\phi^*(g(\theta) + \lambda)] - \lambda\}.$$

Following [33, Section 7.8], let $I_\phi(X;Y) \triangleq \mathrm{D}_\phi(P_{X,Y}||P_X P_Y)$ be the $f$-information, which extends the standard mutual information (MI) defined in terms of KL divergence.

We now set $\theta = (\Delta L_i, U_i)$ and $g(\Delta L_i, U_i) = t G_i$ in Lemma 2.1, where $t \in \mathbb{R}$, $P = P_{\Delta L_i, U_i}$ is the joint distribution of $(\Delta L_i, U_i)$ and $Q = P_{\Delta L_i} P_{U_i}$ is the product of the marginal distributions. Consequently, Lemma 2.1 directly implies that

---

[1]For a function $f : \mathcal{X} \to \mathbb{R} \cup \{-\infty, +\infty\}$, its convex conjugate is $f^*(y) \triangleq \sup_{x \in \text{dom}(f)} \langle x, y \rangle - f(x)$.

$$\mathbb{E}_P\left[G_i\right] \leq \inf_{t \in \mathbb{R}} \frac{1}{t} \left( I_\phi(\Delta L_i; U_i) + \inf_{\lambda \in \mathbb{R}} \left\{ \mathbb{E}_Q\left[\phi^*(t(-1)^{U_i'}\Delta L_i + \lambda)\right] - \lambda \right\} \right) \tag{1}$$

$$\leq \inf_{t \in \mathbb{R}} \frac{1}{t} \left( I_\phi(\Delta L_i; U_i) + \mathbb{E}_Q\left[\phi^*(t(-1)^{U_i'}\Delta L_i)\right] \right), \tag{2}$$

where we let $\lambda = 0$ in Eq. (2). Notice that if $\phi(x) = x\log x + c(x-1)$ for any constant $c$, then $I_\phi(X;Y)$ becomes MI. In this case, the optimal $\lambda^* = -\log\mathbb{E}_Q\left[e^{g(\theta)}\right]$ in Lemma 2.1, and Lemma 2.1 recovers the Donsker and Varadhan's variational formula (cf. Lemma A.1). Here, the second term in Eq. (1) represents the cumulant generating function of $(\Delta L_i, U_i')$. By analyzing the tail behavior of $(\Delta L_i, U_i')$ and applying Eq. (1), one can derive the final MI-based generalization bound. For example, if the loss is bounded, Eq. (1) recovers [24, Theorem 3.2] by using Hoeffding's Lemma.

On one hand, while Eq. (1) is tighter than Eq. (2), the parameter $\lambda$ may not have an analytic form for divergences beyond KL and $\chi^2$. In this sense, Eq. (2) has its own merits. On the other hand, previous MI-based generalization bounds use concentration results to further upper bound the second term in Eq. (1). While this strategy has been successful for the MI case, for instance by using Hoeffding's Lemma, it may be challenging to obtain similar concentration results for other $f$-information. In this work, we introduce a novel approach to prove generalization bounds for $f$-information.

## 3 Conditional $f$-Information Bounds: Bounded Loss Difference Case

We begin by presenting a general lemma that will serve as a main recipe for obtaining generalization bounds in this section. This lemma introduces a new variational formula for $f$-divergence, which may be of independent interest beyond the context of generalization.

**Lemma 3.1.** *Given random variables $X$, $Y$, and a measurable function $f$ for $(X, Y)$. For every $t \in [b_1, b_2] \subseteq \mathbb{R}$, assume that $\phi^*$ is invertible within the range of $t \cdot f$, which we denote by $\Gamma_t$. Let $\phi^{*-1}$ be the inverse of $\phi^*$ on $\cup_{t \in [b_1, b_2]}\Gamma_t$, and let $Y'$ be an independent copy of $Y$. If $\mathbb{E}_{X,Y'}\left[f(X, Y')\right] = 0$, then*

$$\sup_{t \in [b_1, b_2]} \mathbb{E}_{X,Y}\left[\phi^{*-1}(tf(X, Y))\right] \leq I_\phi(X; Y).$$

Although Lemma 3.1 is a simple and straightforward result derived from Lemma 2.1 by setting $g = \phi^{*-1} \circ (tf)$, it is quite powerful for deriving $f$-information-based generalization bounds. Specifically, in the supersample setting, due to the symmetric properties of two columns of $\widetilde{Z}$, we set $X = \Delta L_i$, $Y = U_i$ and $f(\Delta L_i, U_i') = (-1)^{U_i'}\Delta L_i$ (where $U_i'$ is an independent copy of $U_i$), the condition in Lemma 3.1, namely $\mathbb{E}_{X,Y'}\left[f(X, Y')\right] = 0$, is clearly met. Then we have

$$\sup_t \mathbb{E}_{\Delta L_i, U_i}\left[\phi^{*-1}\left(tG_i\right)\right] \leq I_\phi(\Delta L_i; U_i).$$

After carefully choosing $b_1, b_2$ such that $\phi^{*-1}$ is well defined, the main focus is to find a lower bound for the function $\phi^{*-1}$ over a certain interval. We will start with the standard MI case.

### 3.1 Mutual Information (KL-based) Generalization Bound

Consider $\phi(x) = x\log x + x - 1$. Its convex conjugate function is $\phi^*(y) = e^y - 1$ with the inverse $\phi^{*-1}(z) = \log(1 + z)$. Building upon Lemma 3.1, we obtain the following bound:

**Theorem 3.1.** *Assume the loss difference $\ell(w, z_1) - \ell(w, z_2)$ is bounded in $[-1, 1]$ for any $w \in \mathcal{W}$ and $z_1, z_2 \in \mathcal{Z}$, we have*

$$|\mathcal{E}_\mu(\mathcal{A})| \leq \frac{1}{n}\sum_{i=1}^{n} \sqrt{2\left(\mathbb{E}\left[\Delta L_i^2\right] + |\mathbb{E}\left[G_i\right]|\right) I(\Delta L_i; U_i)}.$$

We remark that a bounded loss difference is a less restrictive assumption than a strictly bounded loss. For example, if $\ell$ is $L$-Lipschitz in the second argument, then $\ell(w, z_1) - \ell(w, z_2) \leq L||z_1 - z_2||$.

Thus, as long as $||z_1 - z_2||$ is bounded, the loss difference is bounded even when $\ell$ itself is unbounded. Additionally, we refer to Theorem 3.1 as the "*oracle*" CMI bound because the upper bound itself includes the expected generalization error at position $i$, $\mathbb{E}[G_i]$, which is precisely the quantity we aim to bound.

From Theorem 3.1, first, if we simply upper bound the loss difference $\Delta L_i$ (and $G_i$) by one, Theorem 3.1 recovers [24, Theorem 3.2] up to a constant. More importantly, Theorem 3.1 indicates that in the case of bounded loss difference or bounded loss, solely using $I(\Delta L_i; U_i)$ as the generalization measure for algorithm $\mathcal{A}$ is insufficient to accurately characterize its generalization error. Specifically, when $2\left(\mathbb{E}\left[\Delta L_i^2\right] + |\mathbb{E}[G_i]|\right)$ vanishes as $n$ increases, relying on the MI-based measure alone, i.e., $\sqrt{I(\Delta L_i; U_i)}$, will always result in a slow convergence rate for $|\mathcal{E}_\mu(\mathcal{A})|$. While similar observations have been made in recent studies [41–43], where it is found that the sub-Gaussian variance proxy in previous MI bounds may also vanish in some examples, our Theorem 3.1 provides a more straightforward understanding of the potential looseness of $\mathcal{O}(\sqrt{I(\Delta L_i; U_i)})$. Additionally, further discussion comparing our oracle CMI bound with the MI-based results in [29, 30] is provided in Appendix B.4.

To highlight the differences in our proof techniques compared to previous information-theoretic generalization bounds, we provide a proof sketch below.

*Proof Sketch of Theorem 3.1.* Lemma 3.1 gives us $I(\Delta L_i; U_i) \geq \sup_t \mathbb{E}\left[\log\left(1+t(-1)^{U_i}\Delta L_i\right)\right]$. Let $f(x) = \log(1+x) - x + ax^2$ and set $a = \frac{|\mathbb{E}[G_i]|}{2\mathbb{E}[G_i^2]} + \frac{1}{2}$. By demonstrating that $f(x) \geq 0$ holds when $a \geq \frac{1}{2}$ and $|x| \leq 1 - \frac{1}{2a}$, we have $\sup_{t>-1} \mathbb{E}\left[\log\left(1 + tG_i\right)\right] \geq \sup_{t\in[\frac{1}{2a}-1, 1-\frac{1}{2a}]} \mathbb{E}\left[tG_i - at^2G_i^2\right]$. The supremum is attained when $t^* = \frac{\mathbb{E}[G_i]}{2a\mathbb{E}[G_i^2]} = \frac{\mathbb{E}[G_i]}{|\mathbb{E}[G_i]| + \mathbb{E}[G_i^2]}$, which is achievable. Therefore, $I(\Delta L_i; U_i) \geq \sup_{t\in(-1, +\infty)} \mathbb{E}_{\Delta L_i, U_i}\left[\log\left(1 + t(-1)^{U_i}\Delta L_i\right)\right] \geq \frac{\mathbb{E}^2[G_i]}{4a\mathbb{E}[G_i^2]}$, which simplifies to

$$\mathbb{E}^2[G_i] \leq 2\left(|\mathbb{E}[G_i]| + \mathbb{E}[G_i^2]\right) I(\Delta L_i; U_i). \tag{3}$$

The remaining steps are straightforward. See Appendix B.1 for the complete proof. $\square$

It is also worth noting that by letting $g = \phi^{*-1}$ and $\mathbb{E}_Q[g] = 0$ in the case of KL divergence, we obtain $\lambda^* = 0$ in Lemma 2.1. As a result, Eq. (2) becomes equivalent to Eq. (1). In this sense, starting from Eq. (2) does not compromise the tightness of Eq. (1). Moreover, the proof provides deeper insights into the tightness of Theorem 3.1, which we will now discuss.

**Small Empirical Risk Case or Realizable Setting** Previously, fast-rate information-theoretic generalization bounds in the realizable setting—where the square-root function is removed—have been derived by demonstrating that the CGF can be negative [16, 44, 23, 24], or by invoking the channel capacity result of the binary channel [44, 24].

Based on Theorem 3.1, the square-root function can be directly removed in the realizable setting. Specifically, if the learning algorithm $\mathcal{A}$ is an interpolating algorithm, i.e., the training loss is always minimized to zero, then $G_i \in [0, 1]$ always holds. In this case, $|\mathbb{E}[G_i]| = \mathbb{E}[G_i]$ and $\mathbb{E}[G_i^2] \leq \mathbb{E}[G_i]$, allowing us to obtain $\mathbb{E}[G_i] \leq 4I(\Delta L_i; U_i)$ from Eq. (3). Consequently, Theorem 3.1 simplifies to $\mathcal{E}_\mu(\mathcal{A}) \leq \frac{4}{n}\sum_{i=1}^n I(\Delta L_i; U_i)$. Note that for this bound to hold, the condition of zero training loss can be relaxed to the training loss being always no larger than the testing loss.

Furthermore, to achieve a better constant in the bound, observe that the proof of our Theorem 3.1 utilizes the inequality $\log(1+x) \geq x - ax^2$ for $|x| \leq 1 - \frac{1}{2a}$. If $x \in [0, 1]$ always holds, one can use the inequality $\log(1 + x) \geq x\log 2$ instead, where equality holds when the loss is zero-one loss (See Figure 1(Right) for a visualized illustration).

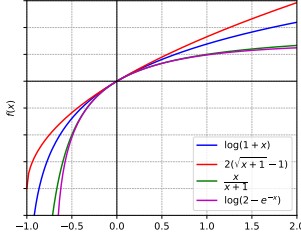 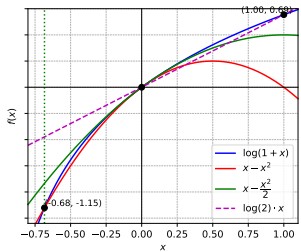

Figure 1: Comparison of different $\phi^{*-1}$ (Left) and examples of $x - ax^2$ for lower-bounding $\log(1+x)$ (Right).

As a result, we obtain $\mathcal{E}_\mu(\mathcal{A}) \leq \sum_{i=1}^n \frac{I(\Delta L_i; U_i)}{n \log 2}$, which is tighter than the previous bound and will never be vacuous (i.e. the bound is always smaller than 1) since $I(\Delta L_i; U_i) \leq H(U_i) = \log 2$. In fact, this bound can exactly characterize the generalization error (i.e., equality holds) if the loss is the zero-one loss, as demonstrated in [24, Theorem 3.3].

**New Fast-Rate Bounds**   Theorem 3.1 not only recovers the previous fast-rate ld-CMI bound in the realizable setting but also introduces new fast-rate bounds.

By solving Eq. (3) for $|\mathbb{E}[G_i]|$, we obtain the following bound.

**Corollary 3.1.** *Under the conditions of Theorem 3.1, we have*

$$|\mathcal{E}_\mu(\mathcal{A})| \leq \frac{1}{n} \sum_{i=1}^n \left( 2I(\Delta L_i; U_i) + \sqrt{2\mathbb{E}[\Delta L_i^2] I(\Delta L_i; U_i)} \right).$$

This fast-rate bound has not been proved in [24], yet it resembles [24, Eq. (2)] in structure. Specifically, the single-loss MI term, $2I(L_i^+; U_i)$ in [24, Eq. (2)], is now substituted with $I(\Delta L_i; U_i)$, and the empirical risk term, $\mathbb{E}[L_S(W)]$, is replaced by $\mathbb{E}[\Delta L_i^2]$. Here, $\Delta L_i = L_i^- - L_i^+$, and notice that both $L_i^+$ and $L_i^-$ follow the same marginal distribution due to the symmetric construction of the supersample, so we can obtain that $\mathbb{E}[\Delta L_i^2] = \mathbb{E}[(L_i^- - L_i^+)^2] \leq 4\mathbb{E}[(L_i^+ - \mathbb{E}[L_i^+])^2] = 4\mathrm{Var}(L_i^+)$. Consequently, Corollary 3.1 further implies the following bound:

$$|\mathcal{E}_\mu(\mathcal{A})| \leq \frac{1}{n} \sum_{i=1}^n \left( 2I(\Delta L_i; U_i) + 2\sqrt{2\mathrm{Var}\left(L_i^+\right) I(\Delta L_i; U_i)} \right). \tag{4}$$

Notably, Eq. (4) hints that if the variance of the single loss in the supersample is sufficiently small, the bound will decay at the same rate as in the realizable setting, namely $\mathcal{O}(I(\Delta L_i; U_i))$, as the second term vanishes.

Instead of solving Eq. (3), if we directly upper-bound $|\mathbb{E}[G_i]|$ in Theorem 3.1, we can also obtain the following bound from Theorem 3.1.

**Corollary 3.2.** *Under the conditions of Theorem 3.1, we have*

$$|\mathcal{E}_\mu(\mathcal{A})| \leq \frac{1}{n} \sum_{i=1}^n \left( \sqrt{2\mathbb{E}[\Delta L_i^2] I(\Delta L_i; U_i)} + \sqrt{2\mathbb{E}_{U_i}\left[\mathrm{D}_{\mathrm{TV}}\left(P_{\Delta L_i|U_i}, P_{\Delta L_i}\right)\right] I(\Delta L_i; U_i)} \right).$$

Note that Corollary 3.2 is tighter than Corollary 3.1 by Jensen's inequality and Pinsker's inequality. While we focus on the MI-based or KL-based generalization bound in this section, the inclusion of total variation (TV) in the second term of Corollary 3.2 expands the scope of the bound. Notably, in a recent study by [34], a component of the form $\sqrt{\mathrm{D}_{\mathrm{TV}}\mathrm{D}_{\mathrm{KL}}}$ is also present in their generalization bounds. Specifically, they derive PAC-Bayesian generalization bounds using the Zhang-Cutkosky-Paschalidis (ZCP) divergence, initially explored in [45]. Interestingly, the ZCP divergence can be further upper bounded by $\mathcal{O}(\sqrt{\mathrm{D}_{\mathrm{TV}}\mathrm{D}_{\mathrm{KL}}} + \mathrm{D}_{\mathrm{TV}})$. Although it remains uncertain whether the first term in Corollary 3.2 is tighter than $\mathrm{D}_{\mathrm{TV}}$ or not, we demonstrate that the component $\sqrt{\mathrm{D}_{\mathrm{TV}}\mathrm{D}_{\mathrm{KL}}}$ can emerge directly in the derivation of KL-based bounds, without using the ZCP divergence. Additionally, note that the term $\mathbb{E}[\Delta L_i^2]$ in Corollary 3.2 can likewise be replaced by $4\mathrm{Var}(L_i^+)$ term.

### 3.2   Other $f$–Information-based Generalization Bound

Based on Lemma 3.1, it is straightforward to apply similar techniques to many well-known $f$-divergences or $f$-information measures to obtain generalization bounds. In this section, we discuss several other $f$-information-based bounds. Before we study the specific $f$-divergence, we remark that, in Section 3.1, our focus was on the unconditional mutual information $I(\Delta L_i; U_i)$. As illustrated in [24], using the data-processing inequality (DPI) and the chain rule, we have $I(\Delta L_i; U_i) \leq I(L_i^+, L_i^-; U_i) \leq I(W; U_i | \widetilde{Z}_i)$. This makes it easy to derive other CMI variant-based bounds. While other $f$-divergences also satisfy DPI, they may not adhere to the chain rule. Accordingly, for these cases, we use the disintegrated conditional $f$-information, defined as $I_\phi^z(X; Y) \triangleq \mathrm{D}_\phi(P_{XY|z} \| P_{X|z} P_{Y|z})$ (noting that $I(X; Y|Z) = \mathbb{E}_Z\left[I_\phi^Z(X; Y)\right]$), instead of the unconditional quantity. In this context, at least $I_\phi(\Delta L_i; U_i | \widetilde{Z}_i) \leq I_\phi(W; U_i | \widetilde{Z}_i)$ still holds, ensuring that the hypothesis-based conditional $f$-information bound can be directly derived.

$\mathcal{X}^2$-**based Generalization Bound**   Consider $\phi(x) = (x-1)^2$. Its convex conjugate is $\phi^*(y) = \frac{y^2}{4} + y$. For $y \geq -2$, the inverse of conjugate function $\phi^{*-1}(z) = 2(\sqrt{z+1} - 1)$ exists. Since $2(\sqrt{z+1}-1) \geq \log(1+z)$, any lower bound that holds for $\log(1+z)$ will also hold for $2(\sqrt{z+1}-1)$. See Figure 1(Left) for a visualization. In other words, all the bounds in Section 3.1 will also hold when using $\chi^2$-divergence. This can also be immediately noticed by the inequality $D_{KL}(P||Q) \leq \chi^2(P||Q)$ [33]. We defer the formal disintegrated $\chi^2$-information bound in Appendix B.7.

**Squared Hellinger (SH) Distance Generalization Bound**   Consider $\phi(x) = (\sqrt{x}-1)^2$, and its convex conjugate is $\phi^*(y) = \frac{y}{1-y}$. For $y \in (-\infty, 1)$, the inverse function $\phi^{*-1}(z) = \frac{z}{1+z}$ exists for $z \in (-1, +\infty)$. We then have the following oracle conditional squared Hellinger (SH)-information bound.

**Theorem 3.2.** *Under the same conditions in Theorem 3.1, we have*

$$|\mathcal{E}_\mu(\mathcal{A})| \leq \frac{1}{n} \sum_{i=1}^n \mathbb{E}_{\widetilde{Z}_i} \sqrt{\left(4\mathbb{E}\left[\Delta L_i^2 | \widetilde{Z}_i\right] + 2\left|\mathbb{E}\left[G_i|\widetilde{Z}_i\right]\right|\right) I_{H^2}^{\widetilde{Z}_i}(\Delta L_i; U_i)},$$

*where $I_{H^2}^{z_i}(\Delta L_i; U_i) = D_{H^2}\left(P_{\Delta L_i, U_i | z_i} || P_{\Delta L_i | z_i} P_{U_i}\right)$ is the (disintegrated) SH-information.*

Given that $D_{H^2}(P||Q) \leq D_{TV}(P, Q)$, the TV-based bound can be derived as a corollary from above.

**Jensen-Shannon (JS) Divergence Generalization Bound**   Consider $\phi(x) = x\log\frac{2x}{1+x} + \log\frac{2}{1+x}$, with the convex conjugate $\phi^*(y) = -\log(2 - e^y)$. For $y < \log 2$, the inverse function $\phi^{*-1}(z) = \log(2 - e^{-z})$ exists for $z > -\log 2$. This leads to the following oracle JS-information bound.

**Theorem 3.3.** *Under the same conditions in Theorem 3.1, we have*

$$|\mathcal{E}_\mu(\mathcal{A})| \leq \frac{2}{n} \sum_{i=1}^n \mathbb{E}_{\widetilde{Z}_i} \sqrt{\left(4\mathbb{E}\left[\Delta L_i^2 | \widetilde{Z}_i\right] + \left|\mathbb{E}\left[G_i|\widetilde{Z}_i\right]\right|\right) I_{JS}^{\widetilde{Z}_i}(\Delta L_i; U_i)},$$

*where $I_{JS}^{z_i}(\Delta L_i; U_i) = D_{JSD}\left(P_{\Delta L_i, U_i | z_i} || P_{\Delta L_i | z_i} P_{U_i}\right)$ is the (disintegrated) JS-information.*

Jeffrey's divergence [46], which is defined as $D_{Jeffrey}(P||Q) \triangleq D_{KL}(P||Q) + D_{KL}(Q||P)$, is an $f$-divergence with the convex function $\phi(x) = (x-1)\log x$. Since $D_{JSD}(P||Q) \leq \frac{1}{4}D_{Jeffrey}(P||Q)$ [47], a Jeffrey's divergence-based bound can be derived as a corollary from Theorem 3.3 (and clearly, also from Theorem 3.1 due to the non-negativity of KL divergence).

Notably, Theorems 3.2-3.3 share similar expressions with Theorem 3.1, and deriving analogous corollaries to Corollaries 3.1-3.2 is feasible (see Appendix B.7).

In the proofs of Theorems 3.2 and Theorem 3.3, we continue to use $x - ax^2$ to lower bound the inverse of conjugate functions $\frac{x}{1+x}$ and $\log(2 - e^{-x})$ for SH-information and JS-information, respectively. However, due to the complexity of handling $\log(2 - e^{-x})$ in the context of JS-information, we opted for a much rougher selection of the parameter $a$ than in the proof of Theorem 3.1 for simplicity. Consequently, the constants in Theorem 3.3 are not optimal and can be refined with a more fine-grained analysis.

## 4   Extension to Unbounded Loss Difference

Previous bounds are typically applicable when dealing with bounded loss differences (albeit not necessarily within the range of $[-1, 1]$). This limitation arises from the fact that the function $\phi^{*-1}$ may fail to exist in Lemma 3.1 when $f$ is unbounded, regardless of any non-trivial adjustments made to the range of $t$. To overcome this limitation, we now extend our analysis to the case of unbounded loss differences by refining Lemma 3.1. One crucial modification to Lemma 3.1 involves defining the measurable function $g = t\phi^{*-1} \cdot \mathbb{1}_{|f| \leq C}$ for some $C \geq 0$, where $\mathbb{1}$ is the indicator function.

We present the following lemma tailored for unbounded functions.

**Lemma 4.1.** *Let $X$ be an arbitrary random variable, $\varepsilon$ be a Rademacher variable, and $t \in (-b, b)$. Assume that there exists a constant $C \geq 0$ such that $\phi^*$ is invertible in $(-bC, bC)$. If $\phi^*(0) = 0$, then*

$$\sup_{t \in (-b,b)} \mathbb{E}_{X,\varepsilon}\left[\phi^{*-1}(t\varepsilon X) \cdot \mathbb{1}_{|X| \leq C}\right] \leq I_\phi(X; \varepsilon).$$

Note that $\phi^*(0) = 0$ is satisfied by all divergences discussed in Section 3. Armed with Lemma 4.1, our proof strategy hinges on separately bounding the truncated terms $\mathbb{E}\left[G_i \cdot \mathbb{1}_{|X| \leq C}\right]$ and $\mathbb{E}\left[G_i \cdot \mathbb{1}_{|X| > C}\right]$. To achieve this, we utilize Lemma 4.1 along with similar lower-bounding techniques as detailed in Section 3 to bound the first term. For the second term, we invoke Hölder's inequality, drawing inspiration from prior works such as [25, 27].

Before presenting the refined bound for unbounded loss difference, we introduce some additional notations. Define the $L_p$ norm of a random variable $X \sim P$ as $||X||_p \triangleq \begin{cases} (\mathbb{E}_P |X|^p)^{\frac{1}{p}} & \text{for } p \in [1, \infty), \\ \text{ess sup } |X| & \text{for } p = \infty, \end{cases}$ where $\text{ess sup } |X| \triangleq \inf\{M : P(|X| > M) = 0\}$ is the essential supremum. Furthermore, let the $f$-divergence generated by the convex function $\phi_\alpha(x) = |x - 1|^\alpha$ be $D_{\phi_\alpha}(P||Q) \triangleq \mathbb{E}_Q\left[\left(\frac{dP}{dQ} - 1\right)^\alpha\right]$. This divergence, which takes $\chi^2$-divergence and total variation as special cases (with $\alpha = 2$ and $\alpha = 1$, respectively), and it has been utilized in previous information-theoretic generalization bounds [25, 29, 30]. We denote the corresponding $f$-information of $D_{\phi_\alpha}(P||Q)$ as $I_{\phi_\alpha}$.

Now, we are ready to present a MI-based bound, with straightforward extensions to other $f$-information measures.

**Theorem 4.1.** *For constants $C \geq 0$, $q \geq 1$, and $\alpha, \beta \in [1, +\infty]$ such that $\frac{1}{\alpha} + \frac{1}{\beta} = 1$, denote $\zeta_1 = \sqrt{2\left(\mathbb{E}\left[\Delta L_i^2 \mathbb{1}_{|\Delta L_i| \leq C}\right] + C\left|\mathbb{E}\left[G_i \mathbb{1}_{|\Delta L_i| \leq C}\right]\right|\right)}$ and $\zeta_2 = (P(|\Delta L_i| > C))^{\frac{q-1}{q\beta}} ||\Delta L_i||_{q\beta}$, we have*

$$|\mathcal{E}_\mu(\mathcal{A})| \leq \inf_{C, q, \alpha, \beta} \frac{1}{n} \sum_{i=1}^{n} \left(\zeta_1 \sqrt{I(\Delta L_i; U_i)} + \zeta_2 \sqrt[\alpha]{I_{\phi_\alpha}(\Delta L_i; U_i)}\right).$$

Since $U_i$ is a Bernoulli random variable, according to [25, Lemma 1], we know that $I_{\phi_\alpha}(\Delta L_i; U_i) < 1 + 2^{\alpha-1}$ for $\alpha \in [1, 2]$ and remains bounded for other values of $\alpha$ as well. Thus, Theorem 4.1 generally preserves the boundedness property of the original CMI bound unless $\Delta L_i$ has an infinite $L_{q\beta}$-norm. Additionally, due to truncation, $\zeta_1 \leq \sqrt{2}C$ for any given $C$, while $\zeta_2$ heavily depends on the tail behavior of $\Delta L_i \sim P_{\Delta L_i}$ (or $G_i \sim P_{U_i'} P_{\Delta L_i}$ equivalently). We now discuss several common cases.

**Bounded Loss** Notably, Theorem 4.1 also covers the bounded loss cases. For instance, if $\ell$ is bounded in $[0, 1]$ a.s., setting $C = 1$ in Theorem 4.1 leads to $\zeta_2 = 0$, directly recovering Theorem 3.1 for bounded loss. In fact, the choice of $C = 1$ might not necessarily be the optimal for $\ell \in [0, 1]$. Particularly, for certain $C \in (0, 1)$, we let $\alpha = 1$ and $\beta = \infty$, then $\zeta_2 = ||\Delta L_i||_\infty$ and $(I_{\phi_\alpha}(\Delta L_i; U_i))^{1/\alpha} = I_{\text{TV}}(\Delta L_i; U_i)$. Since $D_{\text{TV}} \lesssim \sqrt{D_{\text{KL}}}$ by Pinsker's inequality, this alternative choice holds potential for even tighter bounds than Theorem 3.1.

**"Almost Bounded" Loss** When the loss function exhibits sub-Gaussian or sub-Gamma tail behaviors, the probability $P(|G_i| > C)$ decays rapidly. By carefully selecting the threshold $C$, the dominance of the first term in the bound can be expected. Assume the loss difference $\Delta L_i$ is sub-Gaussian with certain variance proxy, say, $\sigma$. Setting $C = \sigma$ and $q = 2$, we have $P(|\Delta L_i| > C) \leq 2e^{-1}$ and $||\Delta L_i||_{q\beta} \lesssim \sqrt{q\beta}\sigma$ [48]. Consequently, each term in the summation of Theorem 4.1 simplifies to $\mathcal{O}\left(\sigma \sqrt{I(\Delta L_i; U_i)} + \sigma(2e)^{\frac{1-\alpha}{2\alpha}} \sqrt{\frac{\alpha}{\alpha-1}} \sqrt[\alpha]{I_{\phi_\alpha}(\Delta L_i; U_i)}\right)$. However, the relationship between $I(\Delta L_i; U_i)$ and $I_{\phi_\alpha}(\Delta L_i; U_i)$ is not clear beyond the cases of $\alpha = 1$ and $\alpha = 2$ (corresponding to total variation and $\chi^2$-information, respectively). Studying the overall behavior of the bound represents an intriguing direction for future research. Furthermore, consider the alternative case with $C = 0$ and $q = 1$, where the first term in Theorem 4.1 becomes zero. The second term, using $||\Delta L_i||_\beta \lesssim \sqrt{\beta}\sigma$ for sub-gaussian random variables, becomes $\mathcal{O}(\sigma \sqrt{\frac{\alpha}{\alpha-1}} \sqrt[\alpha]{I_{\phi_\alpha}(\Delta L_i; U_i)})$. Compared to existing MI bounds, e.g., $\mathcal{O}(\sigma \sqrt{I(\Delta L_i; U_i)})$, we know from $D_{\text{TV}}(P, Q) \lesssim \sqrt{D_{\text{KL}}(P||Q)}$ and $D_{\text{KL}}(P||Q) \leq \chi^2(P||Q)$ that $I_{\phi_1}(\Delta L_i; U_i) \lesssim \sqrt{I(\Delta L_i; U_i)} \lesssim \sqrt{I_{\phi_2}(\Delta L_i; U_i)}$, suggesting that some $\alpha \in (1, 2)$ may yield a tighter bound than MI.

**Heavy-tailed Loss** Heavy-tailed losses, although lacking universally agreed definitions, typically refer to the cases where the moment-generating function (MGF) does not exists (away from 0) [49].

Remarkably, our Theorem 4.1 remains meaningful for such losses since it does not rely on the existence of the MGF of the loss function. Specifically, without any additional knowledge about the tail behavior of $\Delta L_i$, we have the following result from Theorem 4.1.

**Corollary 4.1.** *Under the conditions in Theorem 4.1, let $\gamma = \frac{q-1}{q\beta}$, we have*

$$|\mathcal{E}_\mu(\mathcal{A})| \leq \frac{1}{n} \sum_{i=1}^n \left( \gamma^{\frac{1}{\gamma+1}} + \gamma^{\frac{-\gamma}{\gamma+1}} \right) \left( \sqrt{2I(\Delta L_i; U_i)} \right)^{\frac{\gamma}{\gamma+1}} \left( ||\Delta L_i||_1^\gamma ||\Delta L_i||_{q\beta} \sqrt[\alpha]{I_{\phi_\alpha}(\Delta L_i; U_i)} \right)^{\frac{1}{\gamma+1}}.$$

It's worth noting that as $\gamma \to 0$ (i.e. $q \to 1$), we can see that $\left( \gamma^{\frac{1}{\gamma+1}} + \gamma^{\frac{-\gamma}{\gamma+1}} \right) \to 1$. Consequently, in Corollary 4.1, the bound simplifies to $\frac{1}{n} \sum_{i=1}^n \left( ||\Delta L_i||_\beta \sqrt[\alpha]{I_{\phi_\alpha}(\Delta L_i; U_i)} \right)$. This is an extension of [25, Theorem 3], incorporating individual techniques [6] and loss-difference methods [24] in the supersample setting. Notably, our Corollary 4.1 improves upon [25, Theorem 3], as $\gamma \to 0$ may not necessarily be the optimal choice. Furthermore, it is also intriguing to investigate whether our Corollary 4.1 can recover [29, Corollary 5].

## 5 Numerical Results

In this section, we conduct an empirical comparison between our novel conditional $f$-information generalization bounds and several existing information-theoretic generalization bounds. Our experimental setup closely aligns with the settings in [24]. In particular, we undertake two distinct prediction tasks as follows: 1) *Linear Classifier on Synthetic Gaussian Dataset*, where we train a simple linear classifier using a synthetic Gaussian dataset; 2) *CNN and ResNet-50 on Real-World Datasets*. The second task follows the same deep learning training protocol as in [22], involves training a 4-layer CNN on a binary MNIST dataset ("4 vs 9") [50] and fine-tuning a ResNet-50 model [51], pretrained on ImageNet [52], on CIFAR10 [53]. In both tasks, we assess prediction error as our performance metric. That is, we utilize the zero-one loss function to compute generalization error. Note that during training, we use the cross-entropy loss as a surrogate to enable optimization with gradient-based methods.

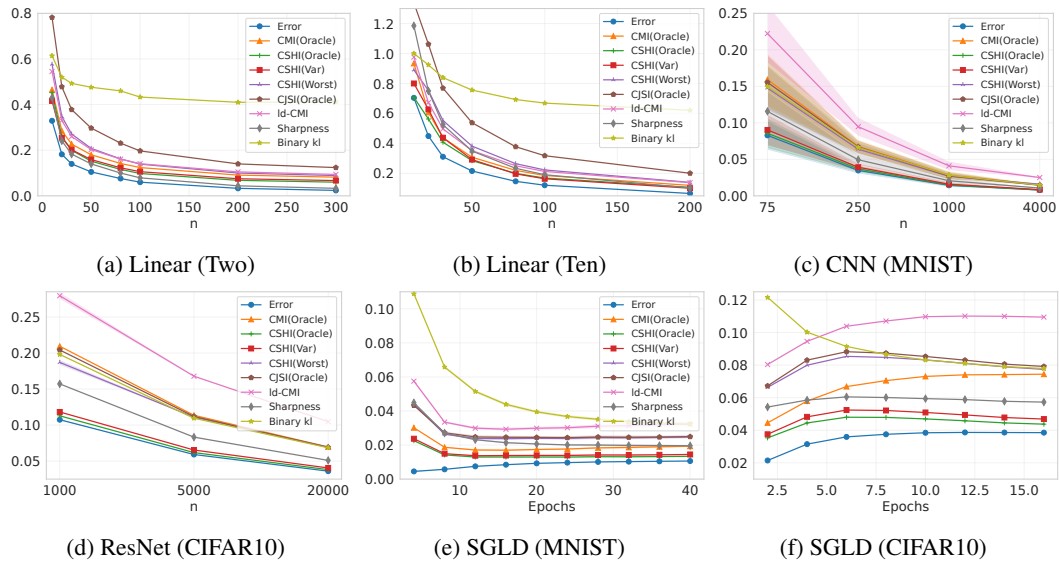

Figure 2: Comparison of bounds on synthetic dataset, MNIST ("4 vs 9"), and CIFAR10. (a-b) Linear classification for two-class and ten-class data. (c-d) Dynamics of generalization bounds as dataset size changes. (e-f) Dynamics of generalization bounds during SGLD training.

The comparison of generalization bounds mainly focuses on those presented in Section 3. Specifically, we consider the oracle bounds, given in Theorem 3.1, Theorem 3.2, and Theorem 3.3, which we denote as *CMI(Oracle)*, *CSHI(Oracle)*, and *CJSI(Oracle)*, respectively. Given that the

squared Hellinger-information is a tighter measure than MI, and the constants in Theorem 3.2 exhibit less pessimism compared to those in the JS-information bound, we introduce two additional squared Hellinger-information bounds for comparison. The first one, akin to Corollary 3.2, is

$$\mathcal{E}_\mu(\mathcal{A}) \le \frac{1}{n} \sum_{i=1}^n \mathbb{E}\sqrt{\left(4\mathbb{E}[\Delta L_i^2] + 2\mathbb{E}\left[\mathrm{D}_{\mathrm{TV}}\left(P_{\Delta L_i|U_i,\widetilde{Z}_i}, P_{\Delta L_i|\widetilde{Z}_i}\right)\right]\right) I_{\mathrm{H}^2}^{\widetilde{Z}_i}(\Delta L_i; U_i)} \text{ (cf. Corol-}$$

lary B.1 in Appendix B.7), denoted as *CSHI(Var)*. The second one adopts a more pessimistic choice by replacing $\mathbb{E}[\Delta L_i^2]$ with the constant 1, labeled as *CSHI(Worst)*. Additionally, we include three previous information-theoretic generalization bounds as baselines: the original ld-CMI bound in [24, Theorem3.1] (*ld-CMI*), the sharpness-based single-loss MI bound in [24, Theorem 4.5] (*Sharpness*), and the binary KL-based CMI bound in [23, Theorem 5] (*Binary kl*). Further details on experimental settings are available in the Appendix D.

The final results are shown in Figure 2. First, we observe that for the linear classifier on the two-class data (Figure 2a), most bounds are close to the exact generalization error, with *Sharpness* being the tightest. Note that the sharpness-based MI bound in [24, Theorem 4.5] is applicable solely for zero-one loss. As the prediction task becomes more complex (Figures 2b-2f), our squared Hellinger-information-based bounds are the tightest. Specifically, the squared Hellinger-information-based bound is always tighter than the CMI bound in the oracle type. Notably, *CSHI(Var)* also outperforms all other $f$-information-based bounds in these cases, and even the pessimistic *CSHI(Worst)* bound outperforms them, except for *Sharpness*, in Figures 2c-2d. Additionally, by comparing *CMI(Oracle)* and *ld-CMI*, we see significant potential for improving the original ld-CMI bound. Finally, although the JS-information bound contains relatively pessimistic constants, it still outperforms some MI-based bounds in many cases (e.g., Figures 2c-2e). These observations suggest that considering alternative $f$-information rather than MI is indeed necessary for studying generalization.

## 6 Other Related Works and Limitations

In addition to the studies mentioned previously, there have been recent developments in information-theoretic generalization bounds in the supersample setting. For example, [54, 44] propose a "leave-one-out" (LOO) construction of supersamples, reducing the size of the supersample from $2n$ to $n + 1$. Additionally, [55] constructs a "neighboring-hypothesis" matrix, enabling the derivation of a hypothesis-conditioned CMI bound. The key component in this bound is the product of the new CMI term and an algorithmic stability parameter. Furthermore, significant progress has been made in understanding MI or CMI as generalization measures in stochastic convex optimization (SCO) problems. Specifically, [56–58] all point out that MI or CMI bounds can be non-vanishing in some SCO examples, while the exact generalization error is vanishing. Notably, [58] discovers a "CMI-accuracy tradeoff" phenomenon in certain scenarios: For the generalization error to be upper bounded by a vanishing parameter, the original hypothesis-based CMI must be lower bounded by the reciprocal of this parameter (or even the reciprocal of its square), indicating that the CMI bound cannot characterize the learnability of such problems. Although this limitation is currently only observed for hypothesis-based CMI, it is anticipated to hold in other variants of CMI, such as ld-CMI.

Our work also contributes to understanding the potential looseness of previous CMI generalization bounds. However, the current state of our bounds cannot explain or resolve the limitation presented in [58] in a non-trivial way. Nonetheless, our framework provides some intuition that, as long as a gap exists between $\phi^{*-1}$ and $x - ax^2$ (see Figure 1(Right)), there will always be room for constructing counterexamples that challenge information-theoretic measures, although such examples may be rare. Another limitation of this work is the lack of a high-probability generalization guarantee. There are two potential directions to address this: either by applying similar techniques to the framework in [59], which is the PAC-Bayesian counterpart of the CMI bound, or by combining it with recent information-bottleneck generalization bounds [60, 61].

## Acknowledgments and Disclosure of Funding

This work is supported partly by an NSERC Discovery grant. The authors would like to thank the anonymous AC and reviewers for their careful reading and valuable suggestions.

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

# Appendices

## A  Technical Lemmas

**Lemma A.1** (Donsker and Varadhan's variational formula). *Let $Q$, $P$ be probability measures on $\Theta$, for any bounded measurable function $f : \Theta \to \mathbb{R}$, we have $\mathrm{D}_{\mathrm{KL}}(Q||P) = \sup_f \mathbb{E}_{\theta \sim Q}[f(\theta)] - \log \mathbb{E}_{\theta \sim P}[\exp f(\theta)]$.*

**Lemma A.2.** *Let $P$ and $Q$ be two probability measures on the same metric space $\mathcal{X}$. Let $f$ be some measurable function associated with $\mathcal{X}$. The total variation distance between $P$ and $Q$ can be written as*

$$\mathrm{D}_{\mathrm{TV}}(P, Q) = \frac{1}{M} \sup_{||f||_\infty \leq M} |\mathbb{E}_P[f(X)] - \mathbb{E}_Q[f(X)]|.$$

**Lemma A.3.** *The following inequalities hold:*

$$\log(1 + x) \geq x - ax^2 \qquad \text{for } a \geq \frac{1}{2} \text{ and } x \in \left[\frac{1}{2a} - 1, 1 - \frac{1}{2a}\right], \tag{5}$$

$$\frac{x}{1 + x} \geq x - ax^2 \qquad \text{for } a \geq 1 \text{ and } x \in \left[\frac{1}{a} - 1, 1 - \frac{1}{a}\right], \tag{6}$$

$$\log(2 - e^{-x}) \geq x - ax^2 \qquad \text{for } a \geq 4 \text{ and } x \in [-\frac{1}{2}, \frac{1}{2}]. \tag{7}$$

*Proof.* We prove these inequalities separately.

**We first prove Eq. (5)**   We analyze the function $f(x) = \log(1 + x) - x + ax^2$. By computing the first derivative $f'(x)$:

$$f'(x) = \frac{1}{1 + x} - 1 + 2ax = \frac{x(2a - 1 + 2ax)}{1 + x}.$$

Let $f'(x) = 0$, we have $x(2a - 1 + 2ax) = 0$, which gives two critical points:

$$x = 0 \quad \text{and} \quad x = \frac{1 - 2a}{2a}.$$

Note that $x = \frac{1-2a}{2a}$ is within the interval $\left[\frac{1}{2a} - 1, 1 - \frac{1}{2a}\right]$. Then computing the second derivative $f''(x)$:

$$f''(x) = \frac{-1}{(1 + x)^2} + 2a.$$

Since $a \geq \frac{1}{2}$ and $-1 < x < 1$, we have $f''(x) \geq 0$, indicating that $f(x)$ is either concave upwards or linear at these critical points. It is easy to check that the minimum value $f(0) = 0$, so $f(x) \geq 0$ in this interval. Hence, $\log(1 + x) \geq x - ax^2$ holds for $a \geq \frac{1}{2}$ and $x \in \left[\frac{1}{2a} - 1, 1 - \frac{1}{2a}\right]$.

**We then prove Eq. (6)**   We analyze the function $f(x) = \frac{x}{1+x} - x + ax^2$. By simplifying this function, we have:

$$f(x) = \frac{x^2(a - 1 + ax)}{1 + x}.$$

Notice that $a - 1 \geq 0$ and $1 - a \leq ax \leq a - 1$, then it is easy to see that the numerator $x^2(a - 1 + ax)$ is always non-negative for $x \in \left[\frac{1}{a} - 1, 1 - \frac{1}{a}\right]$ and $a \geq 1$. In addition, the denominator $1 + x$ is always positive for $x \in \left[\frac{1}{a} - 1, 1 - \frac{1}{a}\right]$. Therefore, $f(x) = \frac{x^2((a-1)+ax)}{1+x} \geq 0$.

Consequently, we have shown that $\frac{x}{1+x} \geq x - ax^2$ for $a \geq 1$ and $x \in \left[\frac{1}{a} - 1, 1 - \frac{1}{a}\right]$.

**We then prove Eq. (7)** We analyze the function $f(x) = \log(2 - e^{-x}) - x + ax^2$. By taking its derivative, we have

$$f'(x) = \frac{1}{2e^x - 1} - 1 + 2ax.$$

We now demonstrate that the function $f'(x) > 0$ for $x \in \left[0, \frac{1}{2}\right]$ and $f'(x) \leq 0$ for $x \in \left[-\frac{1}{2}, 0\right]$.

The second derivative of $f(x)$ is

$$f''(x) = \frac{-2e^x}{(2e^x - 1)^2} + 2a.$$

Let $t = 2e^x$, since $x \in [-\frac{1}{2}, \frac{1}{2}]$, we know that $t \in [2e^{-\frac{1}{2}}, 2e^{\frac{1}{2}}]$. Hence, we reformulate $f''(x)$ as

$$f''(t) = \frac{-t}{(t-1)^2} + 2a = \frac{-1}{t + \frac{1}{t} - 2} + 2a.$$

Notice that $t + \frac{1}{t}$ is monotonically increasing when $t > 1$, so $f''(t)$ is monotonically increasing in $t \in [2e^{-\frac{1}{2}}, 2e^{\frac{1}{2}}]$. Consequently,

$$f''(t) \geq f''(t = 2e^{-\frac{1}{2}}) \approx -26.7 + 2a.$$

Additionally, it is clear that $f''(x = 0) = f''(t = 2e^0) = -2 + 2a > 0$ for $a \geq 4$. WWe now consider two cases:

Case 1. If $a > 13.35$, then $f''(t) \geq f''(t = 2e^{-\frac{1}{2}}) \geq 0$. Hence, $f'(x)$ is monotonically increasing in $x \in [-\frac{1}{2}, \frac{1}{2}]$. Since $f'(x = 0) = 0$, we demonstrate that $f'(x) > 0$ for $x \in \left[0, \frac{1}{2}\right]$ and $f'(x) \leq 0$ for $x \in \left[-\frac{1}{2}, 0\right]$.

Case 2. If $a \leq 13.35$, then there exists an $x_0$ such that $f''(x_0) \leq 0$. Recall that $f''(x = 0) = f''(t = 2e^0) = -2 + 2a > 0$ and $f''(x)$ is monotonically increasing in $x \in [-\frac{1}{2}, \frac{1}{2}]$, we know that such $x_0 \in [-\frac{1}{2}, 0]$ and $f'(x) \geq 0$ for all $x \in \left[0, \frac{1}{2}\right]$. Thus, $f'(x)$ first decreases in $x \in [-\frac{1}{2}, x_0]$ then increases in $x \in [x_0, \frac{1}{2}]$.

Since $f'(x = -\frac{1}{2}) = \frac{1}{2e^{-\frac{1}{2}}} - 1 - a \approx 3.69 - a < 0$ for $a \geq 4$, we conclude that $f'(x) > 0$ for $x \in \left[0, \frac{1}{2}\right]$ and $f'(x) \leq 0$ for $x \in \left[-\frac{1}{2}, 0\right]$.

Consequently, we know that $f(x)$ decreases in $x \in [-\frac{1}{2}, 0]$ and increases in $[0, \frac{1}{2}]$. By checking $f(x) \geq f(0) = 0$, the inequality $\log(2 - e^{-x}) \geq x - ax^2$ holds for $a \geq 4$ and $x \in \left[-\frac{1}{2}, \frac{1}{2}\right]$. This completes the proof. $\square$

# B   Omitted Proofs in Section 3 and Additional Results

## B.1   Proof of Theorem 3.1

*Proof.* Lemma 3.1 indicates that

$$I(\Delta L_i; U_i) \geq \sup_{t \in (-1, +\infty)} \mathbb{E}_{\Delta L_i, U_i} \left[\log\left(1 + t(-1)^{U_i} \Delta L_i\right)\right].$$

Recall that $G_i = (-1)^{U_i} \Delta L_i$, and let $a = \frac{|\mathbb{E}[G_i]|}{2\mathbb{E}[G_i^2]} + \frac{1}{2}$. Since $a \geq \frac{1}{2}$ and $1 - \frac{1}{2a} \geq 0$, we can apply Eq. (5) in Lemma A.3 to have $f(x) = \log(1 + x) - x + ax^2 \geq 0$ for $x \in [\frac{1}{2a} - 1, 1 - \frac{1}{2a}]$.

Hence, by $G_i \in [-1, 1]$, we restrict $t \in [\frac{1}{2a} - 1, 1 - \frac{1}{2a}]$ and let $x = tG_i$ to have the following inequality,

$$\sup_{t > -1} \mathbb{E}\left[\log\left(1 + tG_i\right)\right] \geq \sup_{t \in [\frac{1}{2a} - 1, 1 - \frac{1}{2a}]} \mathbb{E}\left[tG_i - at^2 G_i^2\right] = \frac{\mathbb{E}^2[G_i]}{4a\mathbb{E}[G_i^2]}, \tag{8}$$

where the last equality holds when $t^* = \frac{\mathbb{E}[G_i]}{2a\mathbb{E}[G_i^2]}$.

We now show that $t^*$ can indeed be reached within $t \in [\frac{1}{2a} - 1, 1 - \frac{1}{2a}]$.

Substituting $a = \frac{|\mathbb{E}[G_i]|}{2\mathbb{E}[G_i^2]} + \frac{1}{2}$, we can see that $t^* = \frac{\mathbb{E}[G_i]}{|\mathbb{E}[G_i]| + \mathbb{E}[G_i^2]}$. In addition, notice that

$$1 - \frac{1}{2a} = \frac{|\mathbb{E}[G_i]|}{|\mathbb{E}[G_i]| + \mathbb{E}[G_i^2]} \quad \text{and} \quad \frac{1}{2a} - 1 = \frac{-|\mathbb{E}[G_i]|}{|\mathbb{E}[G_i]| + \mathbb{E}[G_i^2]}.$$

Clearly, either $\mathbb{E}[G_i] = -|\mathbb{E}[G_i]|$ or $\mathbb{E}[G_i] = |\mathbb{E}[G_i]|$ holds. Consequently, $t^*$ can be achieved at one of the endpoints of $[\frac{1}{2a} - 1, 1 - \frac{1}{2a}]$.

Therefore,

$$I(\Delta L_i; U_i) \geq \sup_{t \in (-1, +\infty)} \mathbb{E}_{\Delta L_i, U_i} \left[ \log \left( 1 + t(-1)^{U_i} \Delta L_i \right) \right] \geq \frac{\mathbb{E}^2[G_i]}{4a\mathbb{E}[G_i^2]},$$

which is equivalent to

$$\mathbb{E}^2[G_i] \leq 4a\mathbb{E}[G_i^2] I(\Delta L_i; U_i) = 2 \left( |\mathbb{E}[G_i]| + \mathbb{E}[\Delta L_i^2] \right) I(\Delta L_i; U_i). \tag{9}$$

Then, by Jensen's inequality,

$$\begin{aligned}
\mathcal{E}_\mu(\mathcal{A}) &\leq \frac{1}{n} \sum_{i=1}^{n} |\mathbb{E}_{\Delta L_i, U_i}[G_i]| \\
&\leq \frac{1}{n} \sum_{i=1}^{n} \sqrt{2 \left( |\mathbb{E}[G_i]| + \mathbb{E}[\Delta L_i^2] \right) I(\Delta L_i; U_i)},
\end{aligned}$$

where the last inequality is by Eq. (9).

This completes the proof. $\qquad\qquad\square$

## B.2 Proof of Corollary 3.1

*Proof.* Recall Eq. (3) (or Eq. (9) equivalently),

$$\mathbb{E}^2[G_i] \leq 2 \left( |\mathbb{E}[G_i]| + \mathbb{E}[G_i^2] \right) I(\Delta L_i; U_i).$$

Notice that $|\mathbb{E}[G_i]| \geq 0$, we now solve the inequality above for $|\mathbb{E}[G_i]|$. Let $A_1 = \mathbb{E}[G_i^2]$ and $A_2 = I(\Delta L_i; U_i)$, then by the quadratic-root formula, the solution to $\mathbb{E}^2[G_i] - 2A_2 |\mathbb{E}[G_i]| - 2A_1 A_2 \leq 0$ is

$$0 \leq |\mathbb{E}[G_i]| \leq \frac{2A_2 + \sqrt{4A_2^2 + 8A_1 A_2}}{2} = A_2 + \sqrt{A_2^2 + 2A_1 A_2}.$$

Therefore, we have

$$\begin{aligned}
|\mathbb{E}[G_i]| &\leq I(\Delta L_i; U_i) + \sqrt{\left( I(\Delta L_i; U_i) \right)^2 + 2\mathbb{E}[G_i^2] I(\Delta L_i; U_i)} \\
&\leq 2I(\Delta L_i; U_i) + \sqrt{2\mathbb{E}[G_i^2] I(\Delta L_i; U_i)},
\end{aligned}$$

where the second inequality is by $\sqrt{x+y} \leq \sqrt{x} + \sqrt{y}$.

The remaining steps are routine, which will complete the proof. $\qquad\qquad\square$

## B.3 Proof of Corollary 3.2

*Proof.* Based on Theorem 3.1, by $\sqrt{x+y} \leq \sqrt{x} + \sqrt{y}$, we have

$$|\mathcal{E}_\mu(\mathcal{A})| \leq \frac{1}{n} \sum_{i=1}^{n} \left( \sqrt{2\mathbb{E}[G_i^2] I(\Delta L_i; U_i)} + \sqrt{2|\mathbb{E}[G_i]| I(\Delta L_i; U_i)} \right). \tag{10}$$

Let $G_i' = (-1)^{U_i} \Delta L_i'$, where $\Delta L_i'$ is an independent copy of $\Delta L_i$. Notice that $\mathbb{E}\left[G_i'\right] = 0$. Then,

$$\left|\mathbb{E}_{\Delta L_i, U_i}\left[G_i\right]\right| = \left|\mathbb{E}_{\Delta L_i, U_i}\left[G_i\right] - \mathbb{E}_{\Delta L_i', U_i}\left[G_i'\right]\right| \leq \mathbb{E}_{U_i}\left[\left|\mathbb{E}_{\Delta L_i | U_i}\left[G_i\right] - \mathbb{E}_{\Delta L_i'}\left[G_i'\right]\right|\right],$$

where the last inequality is by Jensen's inequality.

Notice that $G_i \leq 1$, by the dual form of total variation (cf. Lemma A.2), we have

$$\mathbb{E}_{U_i}\left[\left|\mathbb{E}_{\Delta L_i | U_i}\left[G_i\right] - \mathbb{E}_{\Delta L_i'}\left[G_i'\right]\right|\right] \leq \mathbb{E}_{U_i}\left[\mathrm{D_{TV}}\left(P_{\Delta L_i | U_i}, P_{\Delta L_i}\right)\right].$$

Plugging the above into Eq. (10) will complete the proof. $\qquad\square$

## B.4   Comparison with "Online-to-PAC" Generalization Framework

We now discuss the comparison with [29, 30], consider the case of the KL or mutual information. The only explicit *expected* generalization bound provided in [30] is their Corollary 21, which recovers the square-root bound of $\mathcal{O}(\sqrt{I(W;S)/n})$. This bound is clearly weaker than our fast-rate bound in our Corollary 3.1-3.2, due to the omission of vanishing terms in our oracle bound in Theorem 3.1. In fact, a more refined MI bound is presented in the earlier version of [30], namely [29, Corollary 4]. This bound takes the form

$$|\mathcal{E}_\mu(\mathcal{A})| \leq \sqrt{\frac{4\mathbb{E}_Z\|\ell(\cdot, Z) - \mathbb{E}_Z[\ell(\cdot, Z)]\|_\infty^2 I(W;S)}{n}},$$

which can indeed be derived from [30] due to the generality of their framework. Recall that our Theorem 3.1 gives the bound

$$|\mathcal{E}_\mu(\mathcal{A})| \leq \frac{1}{n} \sum_{i=1}^{n} \sqrt{(2\mathbb{E}[\Delta L_i^2] + 2|\mathbb{E}[G_i]|)I(\Delta L_i; U_i)}.$$

Notably, because we apply individual and loss-difference techniques, our averaged MI term is always tighter than that of [29, 30], as $\frac{1}{n}\sum_{i=1}^{n}\sqrt{I(\Delta L_i; U_i)} \leq \sqrt{\frac{I(W;S)}{n}}$ generally holds. To fairly compare our framework with theirs, we ignore the difference between MI terms and only focus on the novel components of each bound, specifically $\mathbb{E}_Z\|\ell(\cdot, Z) - \mathbb{E}_Z[\ell(\cdot, Z)]\|_\infty^2$ in their work and $\mathbb{E}[\Delta L_i^2] + |\mathbb{E}[G_i]|$ in ours, let's consider the following simple example:

**Example 1.** *Let $\mathcal{W} \in [-1, 1]$, and let the input space be $\mathcal{Z} = \{1, -1\}$. Assume $\mu = Unif(\mathcal{Z})$, i.e. $Z$ is a Rademacher variable. Consider a convex and $1$-Lipschitz loss function $\ell(w, z) = -w \cdot z$.*

Under the ERM algorithm, $W = \mathcal{A}(S) = \frac{1}{n}\sum_{i=1}^{n} Z_i$. Notice that for any $w \in \mathcal{W}$, $\mathbb{E}_Z[\ell(w, Z)] = \frac{1}{2}(-w \cdot (1 - 1)) = 0$, hence $\mathbb{E}_Z\|\ell(\cdot, Z) - \mathbb{E}_Z[\ell(\cdot, Z)]\|_\infty^2 = \mathbb{E}_Z\|\ell(\cdot, Z)\|_\infty^2 = 1$. In contrast, since $\Delta L_i \in [-1, 1]$ in this case, $\mathbb{E}[\Delta L_i^2] \leq \mathbb{E}[|\Delta L_i|]$ and $|\mathbb{E}[G_i]| \leq \mathbb{E}[|\Delta L_i|]$, we have $\mathbb{E}[\Delta L_i^2] + |\mathbb{E}[G_i]| \leq 2\mathbb{E}[|\Delta L_i|]$. Moreover, $\mathbb{E}[|\Delta L_i|] = \mathbb{E}[|W \cdot (Z_i^+ - Z_i^-)|] \leq \frac{2}{n}\mathbb{E}[|\sum_{i=1}^{n} Z_i|] \leq \frac{2}{\sqrt{n}}$, where the last step is by the Khintchine-Kahane inequality [62, Theorem D.9].

Thus, in this example, $\mathbb{E}_Z\|\ell(\cdot, Z) - \mathbb{E}_Z[\ell(\cdot, Z)]\|_\infty^2 = 1$, while our bound $\mathbb{E}[\Delta L_i^2] + |\mathbb{E}[G_i]| \leq 2\mathbb{E}[|\Delta L_i|] \leq \mathcal{O}(\frac{1}{\sqrt{n}})$ shows a tighter convergence rate. Consequently, even without using the individual trick and loss-difference technique, our bound (Theorem 3.1) can still be tighter than [29, 30] in terms of convergence rate in this example. This suggests that, MI-based bounds in [29, 30] can still encounter limitations (e.g., slow convergence rate), as with previous MI-based bounds.

Finally, to clarify the comparison, we only demonstrate that our *oracle* CMI bound could be tighter. It is worth noting that the unpublished version of [30] represents an initial exploration of their "online-to-PAC" generalization framework. As such, the results in their current version may primarily aim to recover previous findings, and their framework likely has room for further advancements.

## B.5   Proof of Theorem 3.2

*Proof.* Lemma 3.1 indicates that

$$I_{\mathrm{H}^2}^{\tilde{z}}(\Delta L_i; U_i) \geq \sup_{t \in (-\infty, 1)} \mathbb{E}_{\Delta L_i, U_i | \tilde{z}_i}\left[\frac{t(-1)^{U_i}\Delta L_i}{1 + t(-1)^{U_i}\Delta L_i}\Big| \tilde{z}_i\right].$$

Let $a = \frac{|\mathbb{E}[G_i|\tilde{z}_i]|}{2\mathbb{E}[G_i^2|\tilde{z}_i]} + 1$. Consider the function $f(x) = \frac{x}{1+x} - x + ax^2$. Notice that $a \geq 1$ and $-1 \leq \frac{1}{a} - 1 \leq 1 - \frac{1}{a} \leq 1$. By Eq. (6) in Lemma A.3, we know that when $x \in [\frac{1}{a} - 1, 1 - \frac{1}{a}]$, $f(x) \geq 0$. Hence, recall that $G_i \in [-1, 1]$, we restrict $t \in [\frac{1}{a} - 1, 1 - \frac{1}{a}]$ to have the following inequality,

$$\sup_{t < 1} \mathbb{E}\left[\frac{tG_i}{1 + tG_i}\Big|\tilde{z}_i\right] \geq \sup_{t \in [\frac{1}{a} - 1, 1 - \frac{1}{a}]} \mathbb{E}\left[tG_i - at^2G_i^2|\tilde{z}_i\right] = \frac{\mathbb{E}^2[G_i|\tilde{z}_i]}{4a\mathbb{E}\left[G_i^2|\tilde{z}_i\right]}, \tag{11}$$

where the last equality holds when $t^* = \frac{\mathbb{E}[G_i|\tilde{z}_i]}{2a\mathbb{E}[G_i^2|\tilde{z}_i]}$.

We now show that $t^*$ can indeed be reached within $t \in [\frac{1}{a} - 1, 1 - \frac{1}{a}]$.

Substituting $a$, we can see that

$$t^* = \frac{\mathbb{E}\left[G_i|\tilde{z}_i\right]}{|\mathbb{E}\left[G_i|\tilde{z}_i\right]| + 2\mathbb{E}\left[G_i^2|\tilde{z}_i\right]}.$$

In addition, $1 - \frac{1}{a} = \frac{|\mathbb{E}[G_i|\tilde{z}_i]|}{|\mathbb{E}[G_i|\tilde{z}_i]| + 2\mathbb{E}[G_i^2|\tilde{z}_i]}$ and $\frac{1}{a} - 1 = \frac{-|\mathbb{E}[G_i|\tilde{z}_i]|}{|\mathbb{E}[G_i|\tilde{z}_i]| + 2\mathbb{E}[G_i^2|\tilde{z}_i]}$. Clearly, either $\mathbb{E}\left[G_i|\tilde{z}_i\right] = -|\mathbb{E}\left[G_i|\tilde{z}_i\right]|$ or $\mathbb{E}\left[G_i|\tilde{z}_i\right] = |\mathbb{E}\left[G_i|\tilde{z}_i\right]|$ holds. Consequently, $t^*$ can be achieved in $[\frac{1}{a} - 1, 1 - \frac{1}{a}]$. Therefore,

$$I_{\mathrm{H}^2}^{\tilde{z}}(\Delta L_i; U_i) \geq \sup_{t \in (-\infty, 1)} \mathbb{E}_{\Delta L_i, U_i|\tilde{z}_i}\left[\frac{t(-1)^{U_i}\Delta L_i}{1 + t(-1)^{U_i}\Delta L_i}\Big|\tilde{z}_i\right] \geq \frac{\mathbb{E}^2[G_i|\tilde{z}_i]}{4a\mathbb{E}\left[G_i^2|\tilde{z}_i\right]},$$

which is equivalent to

$$|\mathbb{E}[G_i|\tilde{z}_i]| \leq \sqrt{4a\mathbb{E}\left[G_i^2|\tilde{z}_i\right]I_{\mathrm{H}^2}^{\tilde{z}}(\Delta L_i; U_i)} = \sqrt{\left(2\left|\mathbb{E}\left[G_i|\tilde{z}_i\right]\right| + 4\mathbb{E}\left[G_i^2|\tilde{z}_i\right]\right)I_{\mathrm{H}^2}^{\tilde{z}}(\Delta L_i; U_i)}. \tag{12}$$

Then, by Jensen's inequality,

$$\mathcal{E}_\mu(\mathcal{A}) \leq \frac{1}{n}\sum_{i=1}^n \left|\mathbb{E}_{\Delta L_i, U_i, \tilde{Z}_i}\left[G_i\right]\right| \leq \frac{1}{n}\sum_{i=1}^n \mathbb{E}_{\tilde{Z}_i}\left|\mathbb{E}_{\Delta L_i, U_i|\tilde{Z}_i}\left[G_i\right]\right|.$$

Plugging Eq. (12) into the above will complete the proof. $\qquad\square$

## B.6  Proof of Theorem 3.3

*Proof.* Lemma 3.1 indicates that

$$I_{\mathrm{JS}}^{\tilde{z}}(\Delta L_i; U_i) \geq \sup_{t > -\log 2} \mathbb{E}_{\Delta L_i, U_i|\tilde{z}_i}\left[\log(2 - e^{-t(-1)^{U_i}\Delta L_i})|\tilde{z}_i\right].$$

Let $a = \frac{|\mathbb{E}[G_i|\tilde{z}_i]|}{\mathbb{E}[G_i^2|\tilde{z}_i]} + 4$. Consider the function $f(x) = \log(2 - e^{-x}) - x + ax^2$. Notice that $a \geq 4$, and by Eq. (7) in Lemma A.3, we know that when $x \in [-\frac{1}{2}, \frac{1}{2}]$, $f(x) \geq 0$. Hence, recall that $G_i \in [-1, 1]$, we restrict $t \in [-\frac{1}{2}, \frac{1}{2}]$ to have the following inequality,

$$\sup_{t > -\log 2} \mathbb{E}\left[\log(2 - e^{-tG_i})|\tilde{z}_i\right] \geq \sup_{t \in [-\frac{1}{2}, \frac{1}{2}]} \mathbb{E}\left[tG_i - at^2G_i^2|\tilde{z}_i\right] = \frac{\mathbb{E}^2[G_i|\tilde{z}_i]}{4a\mathbb{E}\left[G_i^2|\tilde{z}_i\right]}, \tag{13}$$

as the same with the previous proofs, the last equality holds when $t^* = \frac{\mathbb{E}[G_i|\tilde{z}_i]}{2a\mathbb{E}[G_i^2|\tilde{z}_i]}$.

We then check the conditions that $t^* \in [-\frac{1}{2}, \frac{1}{2}]$:

$$-\frac{1}{2} \leq \frac{\mathbb{E}\left[G_i|\tilde{z}_i\right]}{2a\mathbb{E}\left[G_i^2|\tilde{z}_i\right]} \leq \frac{1}{2} \qquad \Longleftrightarrow \qquad a \geq \frac{|\mathbb{E}\left[G_i|\tilde{z}_i\right]|}{\mathbb{E}\left[G_i^2|\tilde{z}_i\right]},$$

which is clearly satisfied by $a = \frac{|\mathbb{E}[G_i|\tilde{z}_i]|}{\mathbb{E}[G_i^2|\tilde{z}_i]} + 4$. Consequently, $t^*$ can be achieved in $[-\frac{1}{2}, \frac{1}{2}]$.

Therefore,

$$I_{\mathrm{JS}}^{\tilde{z}}(\Delta L_i; U_i) \geq \sup_{t > -\log 2} \mathbb{E}_{\Delta L_i, U_i | \tilde{z}_i} \left[ \log(2 - e^{-t(-1)^{U_i} \Delta L_i}) | \tilde{z}_i \right] \geq \frac{\mathbb{E}^2[G_i | \tilde{z}_i]}{4a \mathbb{E}\left[G_i^2 | \tilde{z}_i\right]}.$$

With additional arrangements, we have

$$\mathbb{E}[G_i | \tilde{z}_i] \leq \sqrt{4a \mathbb{E}\left[G_i^2 | \tilde{z}_i\right] I_{\mathrm{JS}}^{\tilde{z}}(\Delta L_i; U_i)} = \sqrt{\left(4 \left|\mathbb{E}\left[G_i | \tilde{z}_i\right]\right| + 16 \mathbb{E}\left[G_i^2 | \tilde{z}_i\right]\right) I_{\mathrm{JS}}^{\tilde{z}}(\Delta L_i; U_i)}. \quad (14)$$

Then, by Jensen's inequality,

$$\mathcal{E}_\mu(\mathcal{A}) \leq \frac{1}{n} \sum_{i=1}^n \left| \mathbb{E}_{\Delta L_i, U_i, \widetilde{Z}_i}[G_i] \right| \leq \frac{1}{n} \sum_{i=1}^n \mathbb{E}_{\widetilde{Z}_i} \left| \mathbb{E}_{\Delta L_i, U_i | \widetilde{Z}_i}[G_i] \right|.$$

Plugging Eq. (14) into the above will complete the proof. $\qquad\square$

### B.7 Additional $f$–Information Disintegrated Bounds

The following bounds can be derived by first employing a similar approach as in the proof of Theorem 3.1 to upper bound $\mathbb{E}_{\Delta L_i, U_i | \widetilde{Z}_i}[G_i]$. Then, we incorporate this individual bound into $\mathcal{E}_\mu(\mathcal{A}) \leq \frac{1}{n} \sum_{i=1}^n \left| \mathbb{E}_{\Delta L_i, U_i, \widetilde{Z}_i}[G_i] \right| \leq \frac{1}{n} \sum_{i=1}^n \mathbb{E}_{\widetilde{Z}_i} \left| \mathbb{E}_{\Delta L_i, U_i | \widetilde{Z}_i}[G_i] \right|.$

**Theorem B.1.** *Under the same conditions in Theorem 3.1, we have*

$$|\mathcal{E}_\mu(\mathcal{A})| \leq \frac{1}{n} \sum_{i=1}^n \mathbb{E}_{\widetilde{Z}_i} \sqrt{2 \left( \mathbb{E}\left[\Delta L_i^2 | \widetilde{Z}_i\right] + \left| \mathbb{E}\left[G_i | \widetilde{Z}_i\right] \right| \right) I^{\widetilde{Z}_i}(\Delta L_i; U_i)}.$$

**Theorem B.2.** *Under the same conditions in Theorem 3.1, we have*

$$|\mathcal{E}_\mu(\mathcal{A})| \leq \frac{1}{n} \sum_{i=1}^n \mathbb{E}_{\widetilde{Z}_i} \sqrt{2 \left( \mathbb{E}\left[\Delta L_i^2 | \widetilde{Z}_i\right] + \left| \mathbb{E}\left[G_i | \widetilde{Z}_i\right] \right| \right) I_{\chi^2}^{\widetilde{Z}_i}(\Delta L_i; U_i)},$$

*where $I_{\chi^2}^{\tilde{z}_i}(\Delta L_i; U_i) = \mathrm{D}_{\chi^2}\left(P_{\Delta L_i, U_i | \tilde{z}_i} || P_{\Delta L_i | \tilde{z}_i} P_{U_i}\right)$ is the (disintegrated) $\chi^2$-information.*

The following bound is a corollary from Theorem 3.2, which is used in our experiments in Section 5.

**Corollary B.1.** *Under the same conditions in Theorem 3.1, we have*

$$|\mathcal{E}_\mu(\mathcal{A})| \leq \frac{1}{n} \sum_{i=1}^n \mathbb{E}_{\widetilde{Z}_i} \sqrt{\left( 4\mathbb{E}\left[\Delta L_i^2 | \widetilde{Z}_i\right] + 2\mathbb{E}_{U_i}\left[\mathrm{D}_{\mathrm{TV}}\left(P_{\Delta L_i | U_i, \widetilde{Z}_i}, P_{\Delta L_i | \widetilde{Z}_i}\right)\right] \right) I_{\mathrm{H}^2}^{\widetilde{Z}_i}(\Delta L_i; U_i)}.$$

The proof is nearly the same to Corollary 3.2 except that we do not apply $\sqrt{x + y} \leq \sqrt{x} + \sqrt{y}$ to unpack the terms.

## C Omitted Proof in Section 4

### C.1 Proof of Lemma 4.1

*Proof.* By the variational representation of $f$-divergence, we have

$$\begin{aligned}
I_\phi(X; \varepsilon) &\geq \sup_g \mathbb{E}_{X,\varepsilon}[g(X, \varepsilon)] - \mathbb{E}_{X,\varepsilon'}[\phi^*(g(X, \varepsilon'))] \\
&\geq \sup_t \mathbb{E}_{X,\varepsilon}\left[\phi^{*-1}(t\varepsilon X) \cdot \mathbb{1}_{|X| \leq C}\right] - \mathbb{E}_{X,\varepsilon'}\left[\phi^*\left(\phi^{*-1}(t\varepsilon'X) \cdot \mathbb{1}_{|X| \leq C}\right)\right].
\end{aligned}$$

For the second term above, we have

$$\mathbb{E}_{X,\varepsilon'}\left[\phi^*\left(\phi^{*-1}(t\varepsilon'X) \cdot \mathbb{1}_{|X| \leq C}\right)\right] = \int_\Xi t\varepsilon' X dP_{X,\varepsilon'} + \phi^*(0) = \int_{|x| \leq C} \frac{tX - tX}{2} dP_X = 0,$$

where $\Xi = \{x \in \mathcal{X}, \varepsilon \in \{-1, 1\} | \; |x| \leq C\}$. Putting everything together will conclude the proof. $\quad\square$

## C.2 Proof of Theorem 4.1

*Proof.* Since $(-1)^{U_i}$ is a Rademacher variable and $\phi^*(0) = e^0 - 1 = 0$ for KL divergence, we can apply Lemma 4.1 first then follow the similar developments in Theorem 3.1 by controlling $tG_i \in [-1, 1]$. In particular, for any $C \geq 0$,

$$
\begin{aligned}
I(\Delta L_i; U_i) &\geq \sup_{t \in [-1/C, 1/C]} \mathbb{E}_{\Delta L_i, U_i} \left[ \log(1 + tG_i) \cdot \mathbb{1}_{|G_i| \leq C} \right] \\
&\geq \sup_{t \in [\frac{1-2a}{2aC}, \frac{2a-1}{2aC}]} \mathbb{E}_{\Delta L_i, U_i} \left[ (tG_i - at^2 G_i^2) \cdot \mathbb{1}_{|G_i| \leq C} \right] \\
&= \frac{\mathbb{E}^2[G_i \mathbb{1}_{|G_i| \leq C}]}{4a \mathbb{E} \left[ G_i^2 \mathbb{1}_{|G_i| \leq C} \right]},
\end{aligned}
$$

where $a = \frac{C \left| \mathbb{E}[G_i \mathbb{1}_{|G_i| \leq C}] \right|}{2 \mathbb{E}[G_i^2 \mathbb{1}_{|G_i| \leq C}]} + \frac{1}{2}$. Notice that $t^* = \frac{\mathbb{E}[G_i \mathbb{1}_{|G_i| \leq C}]}{2a \mathbb{E}[G_i^2 \mathbb{1}_{|G_i| \leq C}]} = \frac{\mathbb{E}[G_i \mathbb{1}_{|G_i| \leq C}]}{C \mathbb{E}[G_i \mathbb{1}_{|G_i| \leq C}] + \mathbb{E}[G_i^2 \mathbb{1}_{|G_i| \leq C}]}$
and it is easy to verify $t^* \in [\frac{1-2a}{2aC}, \frac{2a-1}{2aC}]$, so the last equality can be achieved.

Consequently,

$$
\left| \mathbb{E} \left[ G_i \mathbb{1}_{|G_i| \leq C} \right] \right| \leq \sqrt{2(C \left| \mathbb{E} \left[ G_i \mathbb{1}_{|G_i| \leq C} \right] \right| + \mathbb{E} \left[ G_i^2 \mathbb{1}_{|G_i| \leq C} \right]) I(\Delta L_i; U_i)}. \tag{15}
$$

Furthermore, let $P = P_{\Delta L_i, U_i}$ and $Q = P_{\Delta L_i} P_{U_i'}$

$$
\begin{aligned}
\left| \mathbb{E} \left[ G_i \mathbb{1}_{|G_i| > C} \right] \right| &= \left| \int G_i \mathbb{1}_{|G_i| > C} \frac{dP}{dQ} dQ \right| \\
&= \left| \int G_i \mathbb{1}_{|G_i| > C} \left( \frac{dP}{dQ} - 1 \right) dQ \right|,
\end{aligned}
$$

where we use the fact that $\mathbb{E}_{\Delta L_i, U_i'} \left[ G_i \mathbb{1}_{|\Delta L_i| > C} \right] = \frac{\mathbb{E}_{\Delta L_i} \left[ \Delta L_i \mathbb{1}_{|\Delta L_i| > C} - \Delta L_i \mathbb{1}_{|\Delta L_i| > C} \right]}{2} = 0$.
Then, by using Hölder's inequality twice,

$$
\begin{aligned}
\left| \mathbb{E} \left[ G_i \mathbb{1}_{|G_i| > C} \right] \right| &\leq \left( \int |G_i|^\beta \mathbb{1}_{|G_i| > C} dQ \right)^{\frac{1}{\beta}} \left( \int \left( \frac{dP}{dQ} - 1 \right)^\alpha dQ \right)^{\frac{1}{\alpha}} \\
&\leq \left( \int \mathbb{1}_{|\Delta L_i| > C} dQ \right)^{\frac{q-1}{q\beta}} \left( \int |\Delta L_i|^{q\beta} dQ \right)^{\frac{1}{q\beta}} (I_{\phi_\alpha}(\Delta L_i; U_i))^{1/\alpha} \\
&= (P(|\Delta L_i| > C))^{\frac{q-1}{q\beta}} \|\Delta L_i\|_{q\beta} (I_{\phi_\alpha}(\Delta L_i; U_i))^{1/\alpha}. \tag{16}
\end{aligned}
$$

Moreover, notice that

$$
\begin{aligned}
|\mathcal{E}_\mu(\mathcal{A})| \leq \frac{1}{n} \sum_{i=1}^n |\mathbb{E}[G_i]| &= \frac{1}{n} \sum_{i=1}^n \left| \mathbb{E} \left[ G_i \mathbb{1}_{|G_i| > C} + G_i \mathbb{1}_{|G_i| \leq C} \right] \right| \\
&\leq \frac{1}{n} \sum_{i=1}^n \left( \left| \mathbb{E} \left[ G_i \mathbb{1}_{|G_i| > C} \right] \right| + \left| \mathbb{E} \left[ G_i \mathbb{1}_{|G_i| \leq C} \right] \right| \right).
\end{aligned}
$$

Plugging Eq. (15) and Eq. (16) into the above will complete the proof. $\square$

## C.3 Proof of Corollary 4.1

*Proof.* First, by Markov's inequality, we have

$$
\zeta_2 \leq \left( \frac{\mathbb{E}|\Delta L_i|}{C} \right)^\gamma \|\Delta L_i\|_{q\beta} = \frac{\|\Delta L_i\|_1^\gamma \|\Delta L_i\|_{q\beta}}{C^\gamma}.
$$

In addition, recall that $\zeta_1 \leq \sqrt{2} C$, we let

$$
A_1 = \sqrt{2 I(\Delta L_i; U_i)} \quad \text{and} \quad A_2 = \|\Delta L_i\|_1^\gamma \|\Delta L_i\|_{q\beta} \sqrt[\alpha]{I_{\phi_\alpha}(\Delta L_i; U_i)}.
$$

Then, define $f(C) = A_1 C + \frac{A_2}{C^\gamma}$. Solving $\min_{C>0} f(C)$, we have

$$C^* = \left( \frac{A_2 \gamma}{A_1} \right)^{\frac{1}{\gamma+1}} .$$

Plugging $C^*$ into $f(C)$, we have

$$f(C^*) = \left( \gamma^{\frac{1}{\gamma+1}} + \gamma^{\frac{-\gamma}{\gamma+1}} \right) A_1^{\frac{\gamma}{\gamma+1}} A_2^{\frac{1}{\gamma+1}} .$$

From the proof of Theorem 4.1, we notice that

$$\left| \mathbb{E} \left[ G_i \mathbb{1}_{|G_i|>C} \right] \right| + \left| \mathbb{E} \left[ G_i \mathbb{1}_{|G_i|\leq C} \right] \right| \leq f(C^*).$$

Substituting $A_1$ and $A_2$ into the expression of $f(C^*)$, we will finally obtain

$$|\mathcal{E}_\mu(\mathcal{A})| \leq \frac{1}{n} \sum_{i=1}^n \left( \gamma^{\frac{1}{\gamma+1}} + \gamma^{\frac{-\gamma}{\gamma+1}} \right) \left( \sqrt{2I(\Delta L_i; U_i)} \right)^{\frac{\gamma}{\gamma+1}} \left( ||\Delta L_i||_1^\gamma ||\Delta L_i||_{q\beta} \sqrt[\alpha]{I_{\phi_\alpha}(\Delta L_i; U_i)} \right)^{\frac{1}{\gamma+1}} .$$

This completes the proof. $\qquad\square$

## D  Experimental Details

Our experimental setup largely follows [24], and the code of our experiments can be found at https://github.com/ZiqiaoWangGeothe/Conditional-f-Information-Bound.

**Linear Classifier on Synthetic Gaussian Dataset**  In this experiment, similar to [24], we use the popular Python package *scikit-learn* [63] to generate synthetic Gaussian data. Each feature of data is drawn independently from a Gaussian distribution, ensuring that all features are informative for the class labels. Specifically, we set the dimension of feature vector to 5 and create different classes of points, which are normally distributed. We train the linear classifier using full-batch gradient descent with a fixed learning rate of $0.01$ for a total of 300 epochs, employing early stopping when the training error falls below a threshold (e.g., $< 0.5\%$). To ensure robustness and statistical significance, we generate 50 different supersamples for each experiment. Within each supersample, we create 100 different mask random variables, resulting in a total of $5,000$ runs for each experimental setting. This thorough setup allows us to compare and the disintegrated $f$-information-based bounds.

**CNN and ResNet-50 on Real-World Datasets**  In these experiments, the same setup is initially given in [22], and is also used in [23, 24, 61]. Specifically, we draw $k_1$ samples of $\widetilde{Z}$ and $k_2$ samples of $U$ for each given $\tilde{z}$. For the CNN on the binary MNIST dataset, we set $k_1 = 5$ and $k_2 = 30$. The 4-layer CNN model is trained using the Adam optimizer with a learning rate of $0.001$ and a momentum coefficient of $\beta_1 = 0.9$. The training process spans 200 epochs with a batch size of 128. For ResNet-50 on CIFAR10, we set $k_1 = 2$ and $k_2 = 40$. The ResNet model is trained using stochastic gradient descent (SGD) with a learning rate of $0.01$ and a momentum coefficient of $0.9$ for a total of 40 epochs. The batch size for this experiment is set to 64. In the SGLD experiment, we train a 4-layer CNN on the binary MNIST dataset with a batch size of 100 for 40 epochs. The initial learning rate is $0.01$ and decays by a factor of $0.9$ after every 100 iterations. Let $t$ be the iteration index; the inverse temperature of SGLD is given by $\min\{4000, \max\{100, 10e^{t/100}\}\}$. We set the training sample size to $n = 4000$, and $k_1 = 5$ and $k_2 = 30$. Checkpoints are saved every 4 epochs.

All these experiments are conducted using NVIDIA A100 GPUs with 40 GB of memory. For more comprehensive details, including model architectures, we recommend referring to [22–24].

## E  CMI Framework from the Perspective of Coin-Betting Sequences

Deriving generalization bounds from a coin-betting perspective is inspired by [64, 1, 34]. We now adapt this strategy to derive CMI bounds, invoking inequalities similar to those in Lemma A.3.

Assume $\ell \in [0, 1]$. Let $\Delta \ell_i \in [-1, 1]$ be a sequence of "continuous coin" outcomes chosen arbitrarily, and at each round $i$, a player bets $c_i \geq 0$ money on the sign of the outcome $\varepsilon_i \in \{-1, 1\}$. Then, $\Delta \ell_i$ is revealed, and the bettor wins or loses $c_i \varepsilon_i \Delta \ell_i$ money.

Let the initial wealth $\text{Wealth}_0 = 1$. The wealth at the end of round $n$ is:

$$\text{Wealth}_n = \text{Wealth}_{n-1} + c_n \varepsilon_n \Delta \ell_n = 1 + \sum_{i=1}^{n} c_i \varepsilon_i \Delta \ell_i.$$

Notice that $c_i \in [0, \text{Wealth}_{i-1}]$. Now consider a strategy where the player always bets a fixed fraction $t \in [0, 1]$ of the current money, i.e., $c_i = t\text{Wealth}_{i-1}$. Then:

$$\text{Wealth}_n(t) = \text{Wealth}_{n-1} + t\varepsilon_n \Delta \ell_n \text{Wealth}_{n-1} = \prod_{i=1}^{n} (1 + t\varepsilon_i \Delta \ell_i).$$

It follows that

$$\log \text{Wealth}_n(t) = \sum_{i=1}^{n} \log(1 + t\varepsilon_i \Delta \ell_i).$$

Unlike the traditional online gambling game, we incorporate the individual technique from [6] by analyzing the expected log-wealth of the $i$th bet[2], given by

$$W_i(t) = \mathbb{E}_{\varepsilon_i, \Delta L_i} \log \left(1 + t\varepsilon_i \Delta L_i\right).$$

There are two kinds of players: those guessing the coin outcome randomly and those with side information. Let $\varepsilon = (-1)^{U_i}$ where $U_i \sim \text{Bern}(1/2)$. The average log-wealth of the random guesser is $\mathbb{E}_{U_i', \Delta L_i} \log(1 + t(-1)^{U'i} \Delta L_i)$, and the average log-wealth of the informed player is $W_i(t) = \mathbb{E}_{U_i, \Delta L_i} \log(1 + t(-1)_i^U \Delta L_i)$.

Clearly, the value of side information impacts the second player's income. To quantify this impact, by the DV-representation of KL divergence (cf. Lemma A.1), we have

$$I(\Delta L_i; U_i) \geq \sup_{t \in (0,1)} \mathbb{E}_{U_i, \Delta L_i} \log \left(1 + t(-1)^{U_i} \Delta L_i\right) - \log \mathbb{E}_{U_i', \Delta L_i} \left[ e^{\log\left(1 + t(-1)^{U_i'} \Delta L_i\right)} \right]$$

$$= \sup_{t \in (0,1)} \mathbb{E}_{U_i, \Delta L_i} \log \left(1 + t(-1)^{U_i} \Delta L_i\right) - \log \mathbb{E}_{U_i', \Delta L_i} \left(1 + t(-1)^{U_i'} \Delta L_i\right)$$

$$= \sup_{t \in (0,1)} \mathbb{E}_{U_i, \Delta L_i} \log \left(1 + t(-1)^{U_i} \Delta L_i\right), \tag{17}$$

where the last equality holds because $\mathbb{E}_{U_i', \Delta L_i} t(-1)^{U_i'} \Delta L_i = \mathbb{E}_{\Delta L_i} \left[ \mathbb{E}_{U_i'} t(-1)^{U_i'} \Delta L_i \right] = \mathbb{E}_{\Delta L_i} \left[ \frac{t\Delta L_i - t\Delta L_i}{2} \right] = 0$. Notice that the log-wealth of the comparator, i.e. the random-guesser, appears in Eq. (17) in the form of $\inf_\lambda \mathbb{E}_{\Delta L_i, U_i'} \left[ \phi^* \left( \log(1 + t(-1)^{U_i'} \Delta L_i) + \lambda \right) \right] - \lambda$.

Similar to [1], by the inequality $\log(1 + x) \geq x - x^2$ for $x \geq -0.68$ (see Figure 1(Right) for a visualization of this inequality), we have

$$I(\Delta L_i; U_i) \geq \sup_{t \in (0,1)} \mathbb{E}_{U_i, \Delta L_i} \log \left(1 + t(-1)^{U_i} \Delta L_i\right)$$

$$\geq \sup_{t \in (0,0.68)} \mathbb{E}_{U_i, \Delta L_i} \log \left(1 + t(-1)^{U_i} \Delta L_i\right)$$

$$\geq \sup_{t \in (0,0.68)} \mathbb{E}_{U_i, \Delta L_i} \left[ t(-1)^{U_i} \Delta L_i - t^2 \Delta L_i^2 \right]$$

$$\geq \sup_{t \in (0,0.68)} \mathbb{E}_{U_i, \Delta L_i} \left[ t(-1)^{U_i} \Delta L_i - t^2 \right]$$

$$= \frac{1}{4} \mathbb{E}_{U_i, \Delta L_i}^2 \left[ (-1)^{U_i} \Delta L_i \right],$$

We then have

$$\mathbb{E}_{U_i, \Delta L_i} \left[ (-1)^{U_i} \Delta L_i \right] \leq 2\sqrt{I(\Delta L_i; U_i)}. \tag{18}$$

[2]This can also be interpreted as having $n$ players each place a single bet with $t$, so $\log \text{Wealth}_n(t)$ represents their collective log-wealth.

Moreover, we now assume that the side information is "perfect" such that $(-1)^{U_i}\Delta L_i \geq 0$ (i.e. an interpolating or overfitting algorithm) always holds for each $i$, namely the income is always positive. Under this condition, the players can safely set $t = 1$, that is, the players "all-in" their money. Then, by $\log(1 + x) \geq \frac{x}{(1+x)}$ for $x > -1$, we have

$$
\begin{aligned}
I(\Delta_i; U_i) &\geq \mathbb{E}_{U_i, \Delta L_i} \log\left(1 + (-1)^{U_i}\Delta L_i\right) \\
&\geq \mathbb{E}_{U_i, \Delta L_i} \frac{(-1)^{U_i}\Delta L_i}{1 + (-1)^{U_i}\Delta L_i} \\
&\geq \mathbb{E}_{U_i, \Delta L_i} \frac{(-1)^{U_i}\Delta L_i}{2},
\end{aligned}
$$

which gives us

$$
\mathbb{E}_{U_i, \Delta L_i}(-1)^{U_i}\Delta L_i \leq 2I(\Delta_i; U_i). \tag{19}
$$

Consequently, recovering several existing ld-CMI bounds (up to a constant) from Eq. (19) and Eq. (18) is straightforward.

