# OpenReview forum: "Generalization Bounds via Conditional $f$-Information"
_NeurIPS.cc/2024/Conference — NeurIPS 2024 poster_

### Official Review · Reviewer_D8Mx · 2024-06-15

**Soundness:** 3
**Presentation:** 3
**Contribution:** 3
**Rating:** 6
**Confidence:** 3

**Summary:**

This paper introduces novel generalization bounds using the conditional f-information framework. It first derives conditional f-information-based generalization bounds for bounded loss, then examines mutual information generalization bounds as a specific example of their general theorem. This analysis helps to highlight the potential looseness of previous CMI bounds. Additionally, the paper presents several other f-information-based generalization bounds and extends the framework to handle unbounded loss cases. Finally, the tightness of the obtained bounds is demonstrated through empirical comparisons with other existing bounds.

**Strengths:**

- **Clear presentation**: The structure of this paper is clear and easy to follow, with definitions and theorems clearly presented.
- **Empirical validation**: Although it is a theoretical work, the paper includes empirical studies to demonstrate the utility of the proposed generalization bounds.

**Weaknesses:**

As the authors mention, the results pertain to the expected generalization error and therefore lack the high-probability generalization guarantee, which is more critical.

**Questions:**

- For the definition of the expected generalization error (line 101 on page 3), should the correct definition be $\\mathbb{E}\_S[\\mathbb{E}\_W[L_{\\mu}(W)-L\_S(W)]]$ instead of the current formula in the paper? If I understand correctly, $\\mathbb{E}\_W$ refers to taking the expectation with respect to $P\_{W|S}$, which depends on $S$.
- In the “Other Fast-Rate Bound Cases” section, could the authors provide some examples where $\\mathbb{E}[\\Delta L^2\_i] \lesssim \text{Var}(L\_i^{+})$ holds?

**Limitations:**

See weaknesses and questions.

---

> ### Author Rebuttal · Authors · 2024-08-06
>
> We thank you sincerely for the valuable feedback. Our responses follow.
>
> >- For the definition of the expected generalization error (line 101 on page 3), should the correct definition be $\mathbb{E}\_S[\mathbb{E}\_W[L_\mu(W)-L_S(W)]]$ instead of the current formula in the paper? If I understand correctly, $\mathbb{E}\_W$ refers to taking the expectation with respect to $P\_{W|S}$, which depends on $S$.
>
> **Response.** In the definition provided in Line 101, $\mathcal{E}\_\mu(\mathcal{A})\triangleq \mathbb{E}\_W{L\_\mu(W)}-\mathbb{E}\_{W,S}{L\_S(W)}$, the first $\mathbb{E}\_W$ is taken with respect to the marginal distribution $P\_W$. Although $W$ indeed depends on $S$, the population risk $L\_\mu(W)$, as a function of random vairable $W$, only depends on $W$. Therefore, knowing the marginal distribution of $W$ is sufficient to compute $\mathbb{E}\_W{L_\mu(W)}$, namely $\mathbb{E}\_{S,W}{L\_\mu(W)}=\mathbb{E}\_W{L\_\mu(W)}$.
>
>
>
>
>
> >- In the “Other Fast-Rate Bound Cases” section, could the authors provide some examples where $\mathbb{E}[\Delta L^2_i]\lesssim \mathrm{Var}(L_i^+)$ holds?
>
> **Response:** We note that the inequality $\mathbb{E}[\Delta L^2_i]\lesssim \mathrm{Var}(L_i^+)$ holds in the CMI setting in general, rather than being specific to particular examples. This can be demonstrated as follows: In the CMI setting, due to the symmetric nature of the supersample construction, the random variables $L_i^+$ and $L_i^-$ have identical marginal distributions, thus having the same mean $e=\mathbb{E}[L^-_i]=\mathbb{E}[L_i^+]$. Consequently,
>
> $$
> \mathbb{E}[(L^-_i-L_i^+)^2]=\mathbb{E}[(L^-_i-e+e-L_i^+)^2]\leq 2\mathbb{E}[(L^-_i-e)^2]+2\mathbb{E}[(L^+_i-e)^2]=4\mathbb{E}[(L^+_i-\mathbb{E}[L_i^+])^2]=4 \mathrm{Var}(L_i^+).
> $$
>
> Therefore, $\mathbb{E}[\Delta L^2_i]\lesssim \mathrm{Var}(L_i^+)$ is indeed a general result.
>
> We will include this justification in the revision to clarify this point.

---

> > ### Comment · Reviewer_D8Mx · 2024-08-07
> >
> > Thank you for your response to my comments. I have read your rebuttal and will keep my score.

---

> > > ### Author Response · Authors · 2024-08-09
> > > **Thanks**
> > >
> > > Thank you for your prompt reply and for maintaining the positive evaluation of our paper.

---

### Official Review · Reviewer_abMK · 2024-07-04

**Soundness:** 4
**Presentation:** 3
**Contribution:** 3
**Rating:** 7
**Confidence:** 3

**Summary:**

This paper presents a general framework for generalization bounds using a careful application of the Donsker-Vardaran representation of the conditional $f$-information, and show that a suitable quadratic Taylor expansion of the bounds, for various $f$-divergences, improves over the state of the art.

**Strengths:**

This paper has a number of strengths:
(1) The technique appears generic and "fundamental:" it works regardless of $f$ divergence, and unifies the perspectives on quite a few of them. Moreover, it seems to improve over prior art, consistently, and in many instantiations. Finally, Lemma 3.1 does seem useful in its own right. From what I understand, the use of the variational representation also seems novel.
(2) The bound admits extension to unbounded losses, even those which do not admit MGFs
(3) The bounds satisfy an oracle property, in that they adapt to problem-dependent quantities.
(4) For a paper focused on theoretical contribution, the experiments are rather compelling: across toy domains and CIFAR 10, the Hellinger-based oracle confidence interval consistency improves upon prior art.
(5) The authors go to great lengths to explain and differentiate their contributions from prior work.

**Weaknesses:**

I am not an expert in the state-of-art guarantees for generalization, so please take these concerns with a grain of salt. However, my only concern would be that these guarantees do not feel "field changing" in the sense that they appear to be an extension of and elaboration upon an accepted framework: construct a super-sample, evaluate its ($f$-)information, and evaluate the ensuing consequences. The idea of "generalize Shannon information to $f$-information" feels like a natural extension, and therefore its hard for me to find that some of these results are surprising. Moreover, the extension to unbounded losses appears to be based on standard truncation techniques.

That is to say, form an outsider's perspective, this appears a good, solid work with nice fundamental insights. But it doesn't feel groundbreaking enough for me to say, advocate for an award.

There are also a few typos here and there, and the preliminaries read a bit dense at times. This is probably standard in the community, and does not trouble me that much.

Perhaps one suggestion: regarding the presentation, the authors could do well to offer a table in the appendix where they compare the bounds to prior art in a more structured and systematic way. The comparison in the paper often assumes familiarity with prior work, and it would be more effective to re-present the formal statements upon which the authors are improving an an Appendix. Then the authors can more clearly delineate the key advantages of their bounds over past art.

**Questions:**

Had any other works in the literature considered bounds based on $f$-divergences before? Did the original bounds, based on Shannon information, using the variational characterization in any meaningful way? It would be quite helpful for me to understand this better to fully gauge novelty.

**Limitations:**

The authors do an adequate job of explaining the limitations of their work: lack of high probability guarantees, failure to account for the tradeoff in source [52], and perhaps the general limitations of purely information-theoretic approaches to generalization.

---

> ### Author Rebuttal · Authors · 2024-08-06
>
> We thank you sincerely for your valuable feedback and the positive comments on our paper. Our responses follow.
>
> >- I am not an expert in the state-of-art guarantees for generalization, so please take these concerns with a grain of salt. However, my only concern would be that these guarantees do not feel "field changing" in the sense that they appear to be an extension of and elaboration upon an accepted framework: construct a super-sample, evaluate its ($f$-)information, and evaluate the ensuing consequences. The idea of "generalize Shannon information to $f$-information" feels like a natural extension, and therefore its hard for me to find that some of these results are surprising. Moreover, the extension to unbounded losses appears to be based on standard truncation techniques.
>
> >- That is to say, form an outsider's perspective, this appears a good, solid work with nice fundamental insights. But it doesn't feel groundbreaking enough for me to say, advocate for an award.
>
> **Response.** Thank you for considering our paper to be a good and solid work with nice fundamental insights. While we acknowledge that our work builds on existing SOTA information-theoretic generalization bound techniques, such as the CMI framework, our novel change of measure inequalities in Lemma 3.1 and Lemma 4.1 enable the derivation of tighter generalization bounds in a simpler manner. We believe these inequalities can be applied in even broader contexts. Additionally, we sincerely appreciate the reviewer’s evaluation of our paper from an award-level perspective.
>
>
> >- There are also a few typos here and there, and the preliminaries read a bit dense at times. This is probably standard in the community, and does not trouble me that much.
>
> >- Perhaps one suggestion: regarding the presentation, the authors could do well to offer a table in the appendix where they compare the bounds to prior art in a more structured and systematic way. The comparison in the paper often assumes familiarity with prior work, and it would be more effective to re-present the formal statements upon which the authors are improving an an Appendix. Then the authors can more clearly delineate the key advantages of their bounds over past art.
>
> **Response.** Thank you for your valuable suggestions. We will include a table to compare with previous works and put the prior theorem statements in the appendix to improve the readability of our work. In addition, we have fixed some typos and will continue to improve the presentation of our paper in the next revision.
>
> >- Had any other works in the literature considered bounds based on $f$-divergences before? Did the original bounds, based on Shannon information, using the variational characterization in any meaningful way? It would be quite helpful for me to understand this better to fully gauge novelty.
>
> **Response.** Yes, as noted in Lines 43-48, there are previous works that explore using alternative dependence measures, including various $f$-divergences, in place of KL divergence. However, our contribution is distinct in its generic proof approach for $f$-divergence-based bounds under the supersample setting, as detailed in Lemmas 3.1 and 4.1. This method represents a significant departure from previous approaches and provides insight into the tightness of the obtained bound for different $f$-divergences, illustrated by the gap between $\phi^{*-1}(x)$ and $x - ax^2$ in Figure 1. In addition, earlier works typically focus on hypothesis-based $f$-divergence quantities, which are challenging to evaluate empirically.
>
> Regarding the original Shannon information-based bounds, they are indeed based on the Donsker and Varadhan variational formula of KL divergence. These Shannon information-based generalization bounds are obtained by using concentration results (e.g., Hoeffding's Lemma) to upper bound the cumulant generating function in the variational formula, which can be challenging to generalize to other $f$-information. Such challenges are avoided in our Lemmas 3.1 and 4.1, as we discussed in Line 131-136.

---

> > ### Comment · Reviewer_abMK · 2024-08-13
> > **Thank you for the response.**
> >
> > Thank you for the response. I remain positive about the work, and in discussions may recommend a spotlight presentation (if appropriate).

---

> > > ### Author Response · Authors · 2024-08-13
> > > **Thanks**
> > >
> > > We sincerely appreciate your continued positive evaluation of our paper and are grateful for your recommendation for a spotlight presentation.

---

### Official Review · Reviewer_gd7k · 2024-07-15

**Soundness:** 3
**Presentation:** 2
**Contribution:** 2
**Rating:** 6
**Confidence:** 3

**Summary:**

Understanding generalization using information-theoritic measures of dependence between input and output of a learning algorithm is an important area of study. The main focus of this line of work is using the KL divergence as a measure of dependence and providing generalization bounds. In this work, the authors propose a more general approach that replaces the KL divergence with arbitrary f-divergences. The main contribution of this work is presenting an approach to obtain such a generalisation bounds.

**Strengths:**

The main strength of this work is showing KL divergence is not necessary for obtaining information-theoritic generalization bounds. Also, compared to other work showing this fact, the proof strategy in this paper is much simpler.

**Weaknesses:**

I think the weakness of the work are the following:

1- The point that one can replace KL with an arbitrary f-divergence was shown in the following paper:

Lugosi G, Neu G. Online-to-PAC conversions: Generalization bounds via regret analysis. arXiv preprint arXiv:2305.19674. 2023 May 31.

I could not find a clear comparison of the approach proposed in this work and the the paper by Lugosi et.

2- More general question is that by replacing KL with f-divergence what sort of new insights can we obtain? I am trying to understand the real advantage of replacing KL with f-divergences.

3- It is difficult to parse Theorem 4.1. for me. How one can find a good truncation value?

**Questions:**

1- Write down the definition of convex conjugate?
2- One main question here is that what is the gain of using f-divergence compared to the usual mutual information?
3- Typo in Lemma 3.1. It should be b_1 and b_2.

---

> ### Author Rebuttal · Authors · 2024-08-06
>
> We thank you sincerely for your constructive comments on our paper. Our responses follow.
>
> >- 1- The point that one can replace KL with an arbitrary f-divergence was shown in the following paper:
> Lugosi G, Neu G. Online-to-PAC conversions: Generalization bounds via regret analysis. arXiv preprint arXiv:2305.19674. 2023 May 31.
> I could not find a clear comparison of the approach proposed in this work and the the paper by Lugosi et.
>
> **Response.** Our work was indeed inspired by Lugosi et al. (2023) and other recent studies exploring alternative measures of dependency for generalization analysis. While we share a common theme with these works, our proof strategy differs significantly. Notably, our framework is much simpler as it does not rely on existing regret guarantees for online learning algorithms, unlike Lugosi et al. (2023).
>
> Additionally, these works are not directly comparable due to different assumptions (e.g., Lugosi et al. (2023) require the second moment of the loss to be bounded, which we do not). However, both frameworks are deeply connected with convex analysis, and we believe each has its own advantages in specific contexts. We plan to further compare and potentially unify these frameworks in future research.
>
>
> >- 2- More general question is that by replacing KL with f-divergence what sort of new insights can we obtain? I am trying to understand the real advantage of replacing KL with f-divergences.
>
> >- 2- One main question here is that what is the gain of using f-divergence compared to the usual mutual information?
>
> **Response.** Thanks to our generic recipe, we have shown that most generalization bounds based on KL divergence or other $f$-divergence can be derived by lower-bounding $\phi^{\*-1}(x)\geq x-ax^2$. The gap between  $\phi^{\*-1}(x)$ and $x-ax^2$ reflects the tightness of the bound. To elaborate, define the gap function $g(x,a;\phi)=\phi^{\*-1}(x)- (x-ax^2)$. Therefore, to derive a tighter bound (i.e., a smaller $g(x,a;\phi)$), we consider the optimization problem $\min\_{x,a,\phi}g(x,a;\phi)$. Notably, there is no indication that choosing $\phi(x)=x\log(x)$, corresponding to KL divergence, is optimal.
>
> In the paper, we use Figure 1 to visualize this. Clearly, using alternative $f$-divergences such as JS-divergence and squared Hellinger divergence can potentially lead to tighter bounds compared to KL divergence (i.e., the blue line in Figure 1(a)), as they have smaller $g(x,a;\phi)$. This has also been corroborated by our experiments.
>
> While KL divergence and mutual information have advantageous properties like the chain rule, which enable various interesting studies and are not generally applicable to other $f$-divergences, there is no compelling reason to exclusively use KL divergence if tighter bounds can be achieved with other $f$-divergences.
>
>
> >- 3- It is difficult to parse Theorem 4.1. for me. How one can find a good truncation value?
>
> **Response.** We have discussed common cases for selecting truncation values in Lines 295-321. For the bounded case, the truncation value is often chosen as the boundedness value, though it might not always be optimal. If the random variable is likely to stay within a bounded range with high probability, selecting the corresponding boundedness value as the truncation value can be appropriate since $\zeta_2$ will be small in Theorem 4.1. If there is no additional information about the tail behavior of the random variable (e.g., loss difference), we invoke $\zeta_1\leq\sqrt{2}C$, apply the Markov inequality to $\zeta_2$ and then optimize the upper bound with respect to the truncation value $C$. This approach leads to Corollary 4.1.
>
>
> >- 1- Write down the definition of convex conjugate?
>
> **Response.** Thank you for pointing this out. We have added the definition of the convex conjugate to our manuscript.
>
> >- 3- Typo in Lemma 3.1. It should be b_1 and b_2.
>
> **Response.** We appreciate you catching these typos. We have fixed them.

---

> > ### Comment · Reviewer_gd7k · 2024-08-08
> >
> > Thank you for the response. Is there any clean example which shows that the generalization bound developed in your paper is tighter than Lugosi et al. (2023)?

---

> ### Author Response · Authors · 2024-08-09
> **Thanks for the prompt reply.**
>
> The most notable example where our bound is tighter than that of Lugosi et al. (2023) is in the realizable setting, where we provide a strictly non-vacuous bound. Specifically, when $\mathcal{A}$ is an interpolating algorithm (i.e., the training loss is always minimized to zero) and the loss function is bounded in $[0,1]$, as discussed in Lines 197-201, our Lemma 3.1, combined with the lower bound $\log(1+x) \geq x\log{2}$ for $x \in [0,1]$, allows us to derive the bound $\mathcal{E}\_\mu(\mathcal{A}) \leq \sum_{i=1}^n \frac{I(\Delta L_i;U_i)}{n\log{2}}$. This is a strictly non-vacuous bound because $I(\Delta L_i;U_i) \leq \log(2)$, ensuring the overall bound is $\leq 1$, which is the upper bound of the loss function. In contrast, none of the bounds in Lugosi et al. (2023) has this property when the loss is bounded in $[0,1]$; their worst-case bound (e.g., for a deterministic algorithm) exceeds $1$. Additionally, our framework allows similar bounds for other $f$-divergences, e.g., for squared Hellinger distance, using $\frac{x}{1+x} \geq \frac{x}{2}$ for $x \in [0,1]$, a corresponding bound can be obtained.
>
> Moving beyond the realizable setting, consider the case of the KL or mutual information (MI). The only explicit expected generalization bound provided in Lugosi et al. (2023) is their Corollary 21, which recovers the square-root bound of $\mathcal{O}(\sqrt{I(W;S)/n})$. This bound is clearly weaker than our fast-rate bound in Corollaries 3.1-3.2, due to the omission of vanishing terms in our oracle bound in Theorem 3.1. In fact, a more refined MI bound is presented in the earlier version of Lugosi et al. (2023), namely Corollary 4 in Lugosi et al. (2022) [R1]. This bound takes the form $\sqrt{\frac{4\mathbb{E}\_Z\|\|\ell(\cdot,Z)-\mathbb{E}\_Z[\ell(\cdot,Z)]\|\|^2_\infty I(W;S)}{n}}$, which can indeed be derived from Lugosi et al. (2023) due to the generality of their framework. Recall that our Theorem 3.1 gives the bound $\frac{1}{n}\sum_{i=1}^n \sqrt{(2\mathbb{E}[\Delta L^2_i] + 2|\mathbb{E}[G_i]|)I(\Delta L_i;U_i)}$. Notably, because we apply individual and loss-difference techniques, our averaged MI term is always tighter than that of Lugosi et al. (2022, 2023), as $\frac{1}{n}\sum_{i=1}^n \sqrt{I(\Delta L_i;U_i)} \leq \sqrt{\frac{I(W;S)}{n}}$ generally holds. To fairly compare our framework with theirs, we ignore the difference between MI terms and only focus on the novel components of each bound, specifically $\mathbb{E}\_Z\|\|\ell(\cdot,Z)-\mathbb{E}\_Z[\ell(\cdot,Z)]\|\|^2_\infty$ in their work and $\mathbb{E}[\Delta L^2_i]+|\\mathbb{E}[G_i]|$ in ours.
>
> Let’s consider the following simple example:
>
> **Example 1.** Let $\mathcal{W} = [-1,1]$, and let the input space be $\mathcal{Z} = \\{1, -1\\}$. Assume $\mu = \text{Unif}(\mathcal{Z})$, i.e. $Z$ is a Rademacher variable. Consider a convex and 1-Lipschitz loss function $\ell(w,z) = -w \cdot z$.
>
> Under the ERM algorithm, $W = \mathcal{A}(S) = \frac{1}{n}\sum_{i=1}^n Z_i$. Notice that for any $w \in \mathcal{W}$, $\mathbb{E}\_Z[\ell(w,Z)] = \frac{1}{2}(-w \cdot (1-1)) = 0$, hence $\mathbb{E}\_Z\|\|\ell(\cdot,Z) - \mathbb{E}\_Z[\ell(\cdot,Z)]\|\|^2_\infty = \mathbb{E}\_Z\|\|\ell(\cdot,Z)\|\|^2_\infty = 1$. In contrast, since $\Delta L_i \in [-1,1]$ in this case, $\mathbb{E}[\Delta L^2_i] \leq \mathbb{E}[|\Delta L_i|]$ and $|\mathbb{E}[G_i]| \leq \mathbb{E}[|\Delta L_i|]$, we have $\mathbb{E}[\Delta L^2_i] + |\mathbb{E}[G_i]| \leq 2\mathbb{E}[|\Delta L_i|]$. Moreover, $\mathbb{E}[|\Delta L_i|] = \mathbb{E}[|W \cdot (Z_i^+ - Z_i^-)|] \leq \frac{2}{n}\mathbb{E}[|\sum_{i=1}^n Z_i|] \leq \frac{2}{\sqrt{n}}$, where the last step is by the Khintchine-Kahane inequality [R2, Theorem D.9].
>
>
> Thus, in this example, $\mathbb{E}\_Z\|\|\ell(\cdot,Z) - \mathbb{E}\_Z[\ell(\cdot,Z)]\|\|^2_\infty = 1$ as in Lugosi et al. (2022, 2023), while our bound $\mathbb{E}[\Delta L^2_i] + |\mathbb{E}[G_i]| \leq 2\mathbb{E}[|\Delta L_i|] \leq \mathcal{O}(\frac{1}{\sqrt{n}})$ shows a tighter convergence rate. Consequently, even without using the individual trick and loss-difference technique, our bound (Theorem 3.1) is still tighter than Lugosi et al. (2022,2023) in terms of convergence rate. We will include this example in our revision.
>
> Finally, to clarify the comparison, it’s worth noting that the unpublished version of Lugosi et al. (2023) represents an initial exploration of their "online-to-PAC" generalization framework. As such, the results in their current version may primarily aim to recover previous findings, and their framework likely has further potential. We will continue to monitor the development of their work.
>
>
> [R1] G. Lugosi and G. Neu. Generalization Bounds via Convex Analysis. COLT 2022.
>
> [R2] M. Mohri, A. Rostamizadeh, and A. Talwalkar. Foundations of machine learning. MIT press, 2018.

---

> > ### Comment · Reviewer_gd7k · 2024-08-09
> >
> > thank you for the response. I think the authors should update the manuscript and provide a better comparison with Lugosi et al results. I will increase my score to 6.

---

> > > ### Author Response · Authors · 2024-08-10
> > > **Thanks**
> > >
> > > Thank you very much for both your valuable suggestion and the increased score. We will carefully revise our manuscript based on your suggestions.

---

### Official Review · Reviewer_Ka5u · 2024-07-16

**Soundness:** 4
**Presentation:** 3
**Contribution:** 2
**Rating:** 4
**Confidence:** 3

**Summary:**

This paper extends conditional mutual information bounds to other $f$-divergences. A list of bounds involving various $f$-information terms are established. The results are derived by evaluating a previously established variational formula for f-divergences at a specific function. Analysis tailored to various choices of $f$ is then conducted to extract expected generalization bounds.

**Strengths:**

The paper is clear in the assumptions made to derive its results. The buildup to each theorem is well detailed making it easy for the reader to follow.

**Weaknesses:**

The main weakness of this work is in the motivation and the reason for being for the results. The derived extensions have the quantity to be bounded appearing within the bound itself. The leads to strange tautological statements like in line 165 where it is said that if each term in the expected generalization decays fast with $n$ then so does the expected generalization. The $\chi^2$, Hellinger, Jensen-Shannon results have increasingly complicated terms that also involve the very quantity trying to be bound. It is unclear to me why such bounds would be of interest, especially since any instantiation requires falling back to a previously established result.

For example, the sub-gaussian assumption seems to be necessary to have any hope of controlling the quantities appearing in theorem 4.1, and when the assumption is made, the result yields a bound that is worse than existing bounds. The heavy-tail corollary 4.2 is even more impenetrable, the very same quantities we wish to bound appear on the right hand side within $L_p$ norms. The authors should provide justifications as to why these true inequalities are not circular and of limited interest.

Minor note: There is some prior work establishing expected generalization bounds using different $f$-divergences [1].
[1] Esposito, Amedeo Roberto, and Michael Gastpar. "From generalisation error to transportation-cost inequalities and back." 2022 IEEE International Symposium on Information Theory (ISIT). IEEE, 2022.

**Questions:**

How exactly are the experimental plots made ? The theorems involve expectations and mutual information terms, are the authors estimating those? Can the authors add more information on what is plotted?

**Limitations:**

The limitations are discussed.

---

> ### Author Rebuttal · Authors · 2024-08-07
>
> We thank you sincerely for your valuable feedback on our paper. Our responses follow.
>
> >- The main weakness of this work is in the motivation ...
>
> **Response.** As noted in Lines 159-160, when a bound contains $\mathbb{E}[G_i]$, we refer to it as an "oracle" bound, such as Theorem 3.1. Obtaining the oracle bounds first provides new insights into information-theoretic generalization bounds and inspires novel bounds (e.g., Corollaries 3.1-3.2). When the oracle bound falls back to a previous result, our aim is to show either that the previous bound is potentially loose or that it can be recovered by a simpler approach based on our framework.
>
> We now restate our motivation for introducing the $f$-information-based generalization framework. Our framework provides a new, generic method for deriving generalization bounds based on $f$-divergence in the supersample setting, differing from related works by not relying on existing concentration inequalities (e.g., Hoeffding's lemma) or existing regret guarantees for online learning algorithms. The advantages of our framework are twofold: 1) By first obtaining an oracle bound, we show that previous bounds are potentially loose as they ignore some vanishing terms; 2) By carefully handling these vanishing terms, we can either recover previous fast-rate bounds or derive new fast-rate bounds.
>
> To elaborate, consider the KL divergence as an example. Our Theorem 3.1 presents the bound: $\mathcal{E}\_\mu(\mathcal{A})\leq\mathcal{O}(\sqrt{(\mathbb{E}[\Delta L_i^2]+\mathbb{E}[G_i])I(\Delta L_i;U_i)})$. In contrast, existing square-root MI bounds under the same boundedness assumption are $\mathcal{E}_\mu(\mathcal{A})\leq\mathcal{O}(\sqrt{I(\Delta L_i;U_i)})\leq\mathcal{O}(\sqrt{I(W;U_i|\widetilde{Z})})\leq\mathcal{O}(\sqrt{I(W;Z_i)})$. These previous bounds can be very loose as they ignore the vanishing term $\sqrt{\mathbb{E}[\Delta L_i^2]+\mathbb{E}[G_i]}$. Additionally, our framework allows us to recover previous fast-rate bounds such as $\mathcal{O}(I(\Delta L_i;U_i))$ under realizable settings. Furthermore, using the oracle bound, we derive two new fast-rate bounds in Corollaries 3.1-3.2, which do not contain $\mathbb{E}[G_i]$, to mitigate  the looseness in previous square-root bounds.
>
> For other $f$-divergence bounds, we mainly state their "oracle" versions because obtaining similar bounds as in Corollaries 3.1 and 3.2 follows the same procedure. This is mentioned in lines 257-258 (and see Corollary B.1 in Appendix B.6 for the squared Hellinger case).
>
> >- For example, the sub-gaussian assumption ...
>
> **Response.** Regarding the sub-gaussian case, we note that our Theorem 4.1 is not worse than existing bounds. The bound provided in Line 306 uses a rough choice of $C=\sigma$ and a pessimistic upper bound $\zeta_1\leq\sqrt{2}C$. While this bound shows a simple combination of two divergences, it does not rely on the optimal choice of $C$ (which we believe should also vanish as $n$ increases), and $\zeta_1$ is simply replaced by a non-vanishing constant in this case.
>
> In fact, if we want to compare our bound with existing bounds, we can set $C=0$ and $q=1$ in the sub-gaussian case. This makes the first term in Theorem 4.1 zero, and the second term, using $||\Delta L\_i||\_{\beta}\lesssim \beta\sigma$ for sub-gaussian R.V., becomes $\mathcal{O}(\beta\sigma\sqrt[\uproot{5} \alpha]{I\_{\phi_\alpha}(\Delta L_i;U_i)})$. Compared to existing MI bounds, e.g., $\mathcal{O}(\sigma\sqrt{ I(\Delta L_i;U_i)})$, we know from Pinsker's inequality and $\mathrm{KL} \leq \chi^2$ that $\mathcal{O}(I_{\phi_1}(\Delta L_i;U_i))\leq \mathcal{O}(\sqrt{ I(\Delta L_i;U_i)})\leq\mathcal{O}(\sqrt{I_{\phi_2}(\Delta L_i;U_i)})$, suggesting that some $\alpha \in (1,2)$ could outperform the MI bound.
>
> Moreover, even if we set $C = \sigma$ and $q = 1$ as in Line 306, the bound $\mathcal{O}(\sigma\sqrt{ I(\Delta L_i;U_i)}+\frac{\alpha}{\alpha-1}\sigma\sqrt[\uproot{5} \alpha]{I_{\phi_\alpha}(\Delta L_i;U_i)})$ is not necessarily worse than existing bounds in terms of convergence rate. The bound in Line 306 is worse only if $\frac{\alpha}{\alpha-1}\sqrt[\uproot{5} \alpha]{I_{\phi_\alpha}(\Delta L_i;U_i)})$ dominates, namely, if $\frac{\alpha}{\alpha-1}\sqrt[\uproot{5} \alpha]{I_{\phi_\alpha}(\Delta L_i;U_i)})\geq \sqrt{ I(\Delta L_i;U_i)}$. We will add these additional discussions in the revision.
>
> Regarding the heavy-tailed result, there is a typo: $G_i$ in Line 317 should be $\Delta L_i$. We sincerely apologize for the confusion and have fixed it. Furthermore, the validity of our Corollary 4.1 is consistent with existing heavy-tailed generalization bounds, which typically assume the higher $L_p$ norm of the loss is finite, implying the corresponding norm of the loss difference $\Delta L_i$ is finite. Therefore, as long as those bounds are valid, our Corollary 4.1 is meaningful.
>
> >- Minor note: ...
>
> **Response.** Thank you for pointing out this missing reference. We have included in our revision.
>
> >- How exactly are the experimental plots made ...
>
> **Response.** The experimental plots are generated following protocols from previous works on CMI variants [21, 22, 23], and yes, we estimate expectations and MI terms by conducting multiple runs of experiments in the supersample settings. Specifically, as detailed in Appendix D, we draw $k_1$ samples of $\widetilde{Z}$ and $k_2$ samples of $U$ for each given $\tilde{z}$ (e.g., $k_1=5$ and $k_2=30$ for CNN on MNIST). Given that the loss function is 0-1 loss, estimating the mutual information between the discrete random variable $\Delta L_i$ (which can be $0$, $1$, or $-1$) and the binary random variable $U_i$ (which can be $0$ or $1$) is easy. For each plot, we show the mean and standard deviation (represented by the shaded areas) of the estimated bound values and generalization error values. A more detailed description of the experimental protocol is provided in [21, Appendix B], and we will include additional information in the next revision.

---

> ### Comment · Reviewer_Ka5u · 2024-08-13
> **Response to authors**
>
> Thank you for your detailed response.
>
> *Oracle bounds* : I respectfully disagree with the authors on the significance of 'oracle' bounds. The other bounds in the literature are looser exactly because they do not want to include the very terms that need to be bounded in the RHS of the inequality. The authors argument on their benefit remains unfortunately quite unclear to me.
>
> Moreover, the corollaries you mention include expectations of $\Delta L_i^2$ on the right hand side, these are exactly equal to $G_i^2$. This is precisely a generalization gap. Unless I am mistaken, these corollaries are still 'oracle' bounds.
>
> More concerning to me is the following point: the gap $\Delta L_i$ is assumed subgaussian for a fixed $w$'s. The difficulty of bounding the expectations of $\Delta L_i$ lies in the fact that $w$ is data dependent, and therefore it is difficult to control its variance. The authors claims that you could bound them with a constant is very much unclear to me. Why would it be immediate that $\Delta L_i$ is subgaussian, even with learnt weights $w$ ?
>
> *The need for assumptions to make the bounds readable* : I thank the authors for providing details on how one should go about choosing all the different parameters in order to instantiate their bound on a subgaussian loss. I believe this further strengthens the point that the RHS of their results are simply not quantities that are accessible. The motivation for having a generalization bound where the RHS involves $L_p$ norm of the gap $\Delta L_i$ itself is not clear to me.
>
> If the authors could explain why $\Delta L_i^2 = G_i^2$ is a meaningful quantity to have on the RHS, and especially if they could explain why they state that the gap will remain subgaussian even with learnt weights, I would increase my score. Currently, the circularity of the bounds make me inclined to maintain my score.

---

> > ### Author Response · Authors · 2024-08-13
> > **Thanks for the reply**
> >
> > We sincerely appreciate the reviewer's engagement in the discussion.
> >
> > >- Oracle bounds : I respectfully disagree with the authors on the significance of 'oracle' bounds. The other bounds in the literature are looser exactly because they do not want to include the very terms that need to be bounded in the RHS of the inequality. The authors argument on their benefit remains unfortunately quite unclear to me.
> > >- Moreover, the corollaries you mention include expectations of $\Delta L_i^2$ on the right hand side, these are exactly equal to $G_i^2$. This is precisely a generalization gap. Unless I am mistaken, these corollaries are still 'oracle' bounds.
> >
> >
> > **Response.** We believe that stating  $\mathbb{E}[G_i^2]$ is precisely a generalization gap may not be accurate. To our understanding, the generalization gap is $|\mathcal{E}\_\mu(\mathcal{A})|\leq \frac{1}{n}\sum_{i=1}^n|\mathbb{E}[G_i]|=\frac{1}{n}\sum_{i=1}^n\sqrt{\mathbb{E}^2[G_i]}\leq \frac{1}{n}\sum_{i=1}^n\sqrt{\mathbb{E}[G^2_i]}$. Hence, even for a symmetric algorithm $\mathcal{A}$, where $|\mathcal{E}\_\mu(\mathcal{A})|= |\mathbb{E}[G_i]|$, the term $\sqrt{\mathbb{E}[G^2_i]}$ is not exactly a generalization gap.
> >
> > In our paper, we refer to a bound as an 'oracle' bound if it involves $\mathbb{E}[G_i]$. However, if the terms in the bound can be computed solely using $\Delta L_i$, e.g., $\mathbb{E}[\Delta L_i^2]$,  we do not consider it an oracle bound, as the mutual information (MI) term in our bound already involves the random variable $\Delta L_i$. In essence, if the mutual information term requires access to the distribution of $\Delta L_i$, then prohibiting $\Delta L_i^2$ from appearing in the RHS of the bound would imply that the MI term itself may also be disallowed.
> >
> >
> > Furthermore, if one really prefers that $\Delta L_i$ does not appear in the RHS aside from the MI term, we note that in Line 212, we do mention that $\mathbb{E}[\Delta L^2_i]\lesssim \mathrm{Var}[L_i^+]$. This implies that all $\mathbb{E}[\Delta L^2_i]$ terms in these bounds can be replaced by $4\mathrm{Var}[L_i^+]$, which only requires access to a single column of losses in the supersample. This remains a novel bound.
> >
> >
> > Regarding the looseness of other bounds in the literature, we still believe that simply dropping some vanishing terms (i.e., upper-bounding by a constant) does not align with the goal of achieving tight generalization bounds. Instead, explicitly presenting these terms and then carefully handling them is crucial for both understanding why they are loose and improving upon those previous results.
> >
> >
> > >- More concerning to me is the following point: the gap $\Delta L_i$ is assumed subgaussian for a fixed $w$'s. The difficulty of bounding the expectations of $\Delta L_i$ lies in the fact that $w$ is data dependent, and therefore it is difficult to control its variance. The authors claims that you could bound them with a constant is very much unclear to me. Why would it be immediate that $\Delta L_i$ is subgaussian, even with learnt weights $w$ ?
> >
> >
> > **Response.** We appreciate the reviewer raising this valid point, and we acknowledge that our initial argument was indeed unclear. We believe $\Delta L_i$ is subgaussian even with learned weights $w$ for the following reasons: $L_i^-$ or $L_i^+$ can either be training loss (i.e., $W$ depends on $Z$) or testing loss (i.e., $W$ is independent of $Z$). If $L_i^-$ is a testing loss, it is subgaussian, and if $L_i^-$ is a training loss, we rely on the common understanding that the training loss should be finite for any meaningful algorithm (i.e., bounded by some constant), making it subgaussian as well. Therefore, the overall loss difference, being the sum of two subgaussian random variables, remains subgaussian. That said, we acknowledge that the additional condition—that the training loss should not go to infinity—should be explicitly stated. We will revise Lines 303-309 to directly consider $\Delta L_i$ as $\sigma$-subgaussian (rather than for any fixed $w$) and incorporate the related discussions in our previous response.
> >
> > >- The motivation for having a generalization bound where the RHS involves $L_p$ norm of the gap $\Delta L_i$ itself is not clear to me.
> >
> > **Response.** We maintain the opinion that if the mutual information term is allowed to be defined based on $\Delta L_i$, then involving the $L_p$ norm of the gap $\Delta L_i$ should also be valid. Moreover, it is still possible to replace this with the $L_p$ norm of the (centered) single-column loss due to the symmetric property of the supersample construction.
> >
> >
> >
> > Please do let us know if the reviewer has any remaining concerns about the motivation behind our work.

---

### Decision · Program_Chairs · 2024-09-25

**Decision:**

Accept (poster)

**Comment:**

This paper introduces a general framework for deriving generalization bounds using a careful application of the Donsker-Varadhan representation. The reviewers found the approach solid and the contributions meaningful. I recommend acceptance.